# Tomographic detection of photon pairs produced from high-energy X-rays for the monitoring of radiotherapy dosing

**Qihui Lyu ⓘ , Ryan Neph and Ke Sheng ⓘ ✉**

Measuring the radiation dose reaching a patient's body is difficult. Here we report a technique for the tomographic reconstruction of the location of photon pairs originating from the annihilation of positron–electron pairs produced by high-energy X-rays travelling through tissue. We used Monte Carlo simulations on pre-recorded data from tissue-mimicking phantoms and from a patient with a brain tumour to show the feasibility of this imaging modality, which we named 'pair-production tomography', for the monitoring of radiotherapy dosing. We simulated three image-reconstruction methods, one applicable to a pencil X-ray beam scanning through a region of interest, and two applicable to the excitation of tissue volumes via broad beams (with temporal resolution sufficient to identify coincident photon pairs via filtered back projection, or with higher temporal resolution sufficient for the estimation of a photon's time-of-flight). In addition to the monitoring of radiotherapy dosing, we show that image contrast resulting from pair-production tomography is highly proportional to the material's atomic number. The technique may thus also allow for element mapping and for soft-tissue differentiation.

There is a long history of using X-rays for detection[1]. Besides industrial inspection[2], X-rays hold a uniquely important position in medicine[3]. Compared with other medical imaging techniques such as magnetic resonance imaging (MRI)[4–6] and positron emission tomography (PET)[7], X-ray imaging systems are advantageous in their low cost, high speed, high resolution and high sensitivity to dense or high-atomic-number materials. A major breakthrough in X-ray imaging was the invention of tomographic images[8,9]. By acquiring two-dimensional (2D) X-ray attenuation images from many different angles around the patient, a 3D computed tomography (CT)[10,11] image can be reconstructed[8,12]. CT revolutionized modern medicine by enabling diagnoses that were previously simply impossible.

Since the invention of CT, many technological developments in the hardware and reconstruction methods have markedly improved its speed, image quality and versatility[13–15]. Nonetheless, the underlying mechanism for image formation remains the same. X-ray CT imaging signals are produced by a mixture of physical interactions, including Rayleigh scatter, photoelectric effect and Compton scatter. These fundamentally different interactions result in different attenuation patterns with regard to the material properties. The problem is further complicated by the bremsstrahlung poly-energetic X-rays commonly available for diagnosis and therapy. As a result, it is difficult to obtain clean material information from the X-ray CT images. Dual-energy and photon-counting CT help improve material differentiation, but they are limited in the number of differentiable basis materials and do not directly quantify material atomic numbers[16]. Another weakness of CT is poor soft-tissue contrast due to the similarity in densities and the diminishing photoelectric effect with low-atomic-number materials. Phase-contrast CT can improve soft-tissue contrast, but it requires either coherent X-ray sources or additional optical elements such as Talbot-Lau gratings. Owing to these additional complexities, significant research and development are still needed to make phase-contrast CT a clinically viable modality[17–19]. For tomographic reconstruction, attenuation signals from sufficient X-ray beam angles are required.

Department of Radiation Oncology, University of California Los Angeles, Los Angeles, CA, USA. ✉e-mail: ksheng@mednet.ucla.edu

In specific cases, with machine learning[20] and sparse regularization[21] methods, the requirement can be relaxed to predict useable images, but a general solution for sparse-view tomographic image reconstruction does not exist[22]. Furthermore, CT reconstruction based on the Radon transform has global support, meaning that truncated projections with a partial view of the patient would inevitably introduce inaccuracies whose magnitude depends on the degree of truncation and reconstruction method. One undesired consequence of the data sufficiency requirement is the exposure of a large patient volume to the imaging dose regardless of the size of the region of interest (ROI).

Another major application of X-rays in modern medicine is radiotherapy of cancer. The ionizing radiation from high-energy X-rays can break DNA strands which, if not repaired, can lead to cell death. By exploiting the differential repair mechanisms of cancer and normal cells, and the additional therapeutic contrast due to conformal dose distribution, radiotherapy has been a mainstay modality in cancer treatment. It is estimated that 60% of cancer patients and 40% of the curative cases in the United States use radiotherapy as either one of or the only treatment method[23].

X-ray-based radiotherapy is an open-loop treatment, meaning that the delivered 3D dose in the patient is not directly verified. Compared with a closed-loop system, an open-loop system is intrinsically less safe and less accurate due to the lack of direct feedback. In vivo radiation dose deep inside the patient's body is difficult to measure. Implanted dosimeters require an undesirable interventional procedure and still only measure point doses[24]. Cerenkov imaging is limited to superficial locations[25]. X-ray induced acoustic CT (XACT) has shown promise to measure 3D in vivo dosimetry. However, XACT applications are hampered by the acoustic boundaries, low resolution and signal-to-noise ratio (SNR), and mandatory ultrasound receiver arrays that interfere with the X-ray beam path[26]. Because of the fundamental impediments, these in vivo dosimetry methods are unlikely to meet the general needs of 3D in vivo dosimetry to close the radiotherapy open loop.

To meet the challenges and expand the applicability of X-ray tomography, we introduce a distinctly new 3D X-ray image-formation method: pair-production tomography (P2T) imaging. P2T is similar to PET in terms of how imaging is done: they both measure coincident annihilation photons emitted from positron annihilation. The only difference is the source of positrons: PET introduces positrons with radioactive tracers, while P2T introduces positrons through mega-voltage X-ray-induced pair production. P2T is similar to CT in terms of what it is imaging: the image signals in both depend on the material composition. They differ in image-formation mechanisms: CT measures X-ray transmission, while P2T measures pair-production signals. With its unique image-formation method, we show that P2T provides a direct verification of 3D radiotherapy dose. It also serves as an imaging modality with a contrast different from that of CT, provides a clean linear relationship to the material atomic number, and P2T images can be formed even with partial-view and sparse-view projections. A previous study used pair production for one-sided point material detection[27], where the radiation source and a single detector module are located on the same side of the object, and only a single point can be measured at a time. Different from the previous work, we use coincidence information to form 3D P2T images, which substantially expands the capacity for medical applications.

## Principles of pair-production tomography imaging

Figure 1a illustrates the major X-ray interactions in the P2T energy range (for example, 10 MV bremsstrahlung source, which is commonly used in radiotherapy). In the photoelectric effect, an incident photon vanishes after striking a bound electron, resulting in the ejection of the electron and a vacancy in the inner shell. To stabilize the atom, an outer shell electron fills the vacancy and converts the energy lost to characteristic radiation X-ray or an Auger electron. In Compton scattering, an incident photon is scattered by a charged particle, typically an electron, and transfers part of the photon energy to the recoiling electron. The pair production occurs in a Coulomb force field, typically near a nucleus, where an incident X-ray of sufficiently high energy (at least 1.022 MeV) is annihilated and produces a positron and an electron. Subsequently, the electron dissipates energy through successive interactions with the medium before being absorbed by the medium. However, as the positron loses its kinetic energy and comes to a near stop, it encounters an electron with nearly simultaneous annihilation of the positron and the electron, and their conversion into two annihilation photons moving in opposite directions with an energy of around 511 keV. The pair-production attenuation coefficient is linear to the material atomic number[28].

The P2T formation process is illustrated in Fig. 1b. An X-ray beam typically used for radiotherapy introduces the pair-production electron-positron pair in a subject placed at the centre of a ring-detector array. Before producing the two time-coincident 511 keV annihilation photons, the positron would travel for a median distance of 4.6 mm for a 10 MV beam (Supplementary Fig. 1). The two annihilation photons travel in opposite directions, captured by two detectors on the ring. A 3D map of the event locations can then be reconstructed on the basis of a collection of the signals.

The photon contamination from photoelectric and Compton interactions is effectively reduced via a coincidence time window and an energy window. Figure 1c shows the energy distribution of detected photons ranging from 0 to 1 MeV, with a zoom-in view of 0.511 MeV ± 10% energy range (Fig. 1d,e). The 511 keV photons comprise 0.91% of the total photons before applying filters (Fig. 1c), 15.4% after applying a ±10% energy window filter (Fig. 1d) and 77.6% after applying the ±10% energy as well as a 1 ns coincidence time filter (Fig. 1e).

Two P2T excitation approaches are investigated in this study (Fig. 1f). The volume excitation (VE) approach excites the entire imaging field simultaneously at each view angle. In scanning pencil beam excitation (SPBE), the imaging field is excited sequentially. Note that the imaging field can be full view or partial view of the imaged object. Full-view imaging covers the entire imaged object, while partial-view imaging only irradiates the ROI without exposing the majority of the imaged object. SPBE affords additional geometrical information for tomographic reconstruction by pinpointing the pair-production location at the intersection of the pencil beam and the detector coincidence line.

## Simulation, reconstruction and post processing

A general-purpose Monte Carlo (MC) package, Geant4 (ref. [29]), was used to characterize P2T. A ring-detector array with a total of 1,440 detector elements and a diameter of 240 cm were assumed. For simplicity, we set the ring detector to have only a single row with 10 cm width in the z-direction (for example, patient superior-inferior direction), which covers 4.17% of the solid angles. A ±10% energy window was assumed. The photon detection time can be computed as the sum of the photon releasing time, the photon travelling time and the detector response time. The primary photons within each pencil beam are released in sequence, with time intervals following a uniform distribution. The detector response time was simulated as a Gaussian distribution with a standard deviation equal to the time resolution of the detector.

We studied three imaging acquisition and reconstruction methods: filtered back projection (FBP), scanning pencil beam (SPB) and time of flight (TOF). Coincident events were identified as two energy-eligible photons (within the energy window) arriving at two detector elements within the coincidence time of 1 ns. Two coincident events define a line of response (LOR): the line connecting the two detector elements, indicating that the annihilation event happened on the LOR. Once LORs are identified, they are re-binned to sinogram and then reconstructed using FBP with the Michigan Image Reconstruction Toolbox (MIRT)[30] (the FBP method). The SPB imaging method uses the scanning pencil beam excitation method and excites the imaging

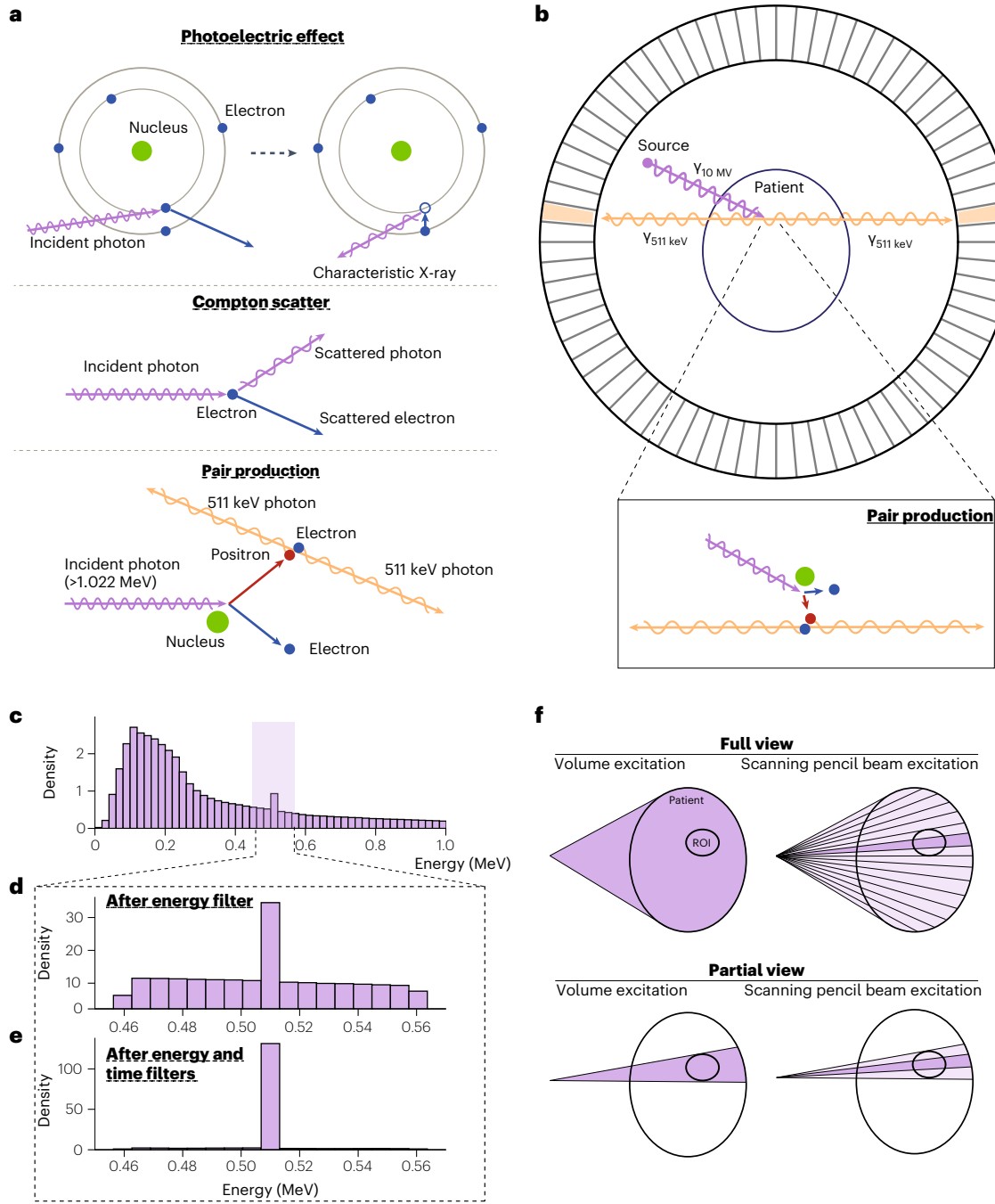

**Fig. 1 | Illustration of the principles of pair-production tomography imaging.**
**a**, Illustration of the photoelectric effect, Compton scatter and pair-production interaction. **b**, Illustration of the formation process of pair-production tomography imaging (P2T). **c**, The energy distribution of detected photons ranging from 0 to 1 MeV. **d**, The energy distribution of detected photons after applying a ±10% energy window filter. **e**, The energy distribution of detected

photons after applying the ±10% energy and 1 ns coincidence time filters.
**f**, Comparison of VE vs SPBE, and full-view imaging (top) vs partial-view imaging (bottom). VE excites the entire imaging field simultaneously at each view angle. In SPBE, the imaging field is excited sequentially. The partial-view imaging only irradiates the ROI without exposing the majority of the imaging object.

ROI sequentially using thin pencil beams. With the known excitation path, SPB locates each annihilation event as the intersection of the corresponding LOR and the pencil beam path (the SPB method). For detectors with a high time resolution (such as 20 ps), the range of the annihilation event along the LOR can be narrowed down (to, for example, 3 mm), according to the time difference of the two photons (the TOF method). In addition to the reconstructed images, the ground-truth (GT) image was created as the voxel-wise tally of positron annihilation events.

We considered two different detector time resolutions: 20 ps or 300 ps, representing the upper limits of experimental Cerenkov[31,32] and state-of-the-art commercial scintillator detectors[33], respectively. The TOF information of the 20 ps detector allows directly locating an annihilation event with 3 mm resolution. A TOF of 300 ps results in a 45 mm range, which in itself is inadequate resolution. However, when a large number of annihilation events are recorded, such as in high-dose radiotherapy, useful images can be reconstructed in a statistical manner using FBP. SPB increases resolution to 2 mm by using a

**Table 1 | Summary of the excitation approaches, the assumed detector time resolution and correction methods**

| | | Imaging acquisition and reconstruction methods | | | |
| --- | --- | --- | --- | --- | --- |
| | | **GT** | **FBP** | **SPB** | **TOF** |
| Excitation approaches | | Both | VE | SPBE | VE |
| Detector time resolution | | NA | 300 ps | 300 ps | 20 ps |
| Radiotherapy | Attenuation correction | No | Yes | Yes | Yes |
| | Fluence correction | No | No | No | No |
| Quantitative imaging | Attenuation correction | No | Yes | Yes | Yes |
| | Fluence correction | Yes | Yes | Yes | Yes |

The imaging methods include GT, FBP, SPB and TOF. The excitation approaches include SPBE and VE. The GT images are the same for both SPBE and VE. NA, not applicable.

2 mm excitation beam even with a 300 ps detector. Therefore, we only considered TOF reconstruction for a 20 ps detector and SPB or FBP for a 300 ps detector in this study.

Similar to PET, the attenuation to annihilation photons is compensated for by weighting the coincident photon-pair counting on the basis of their respective radiological path lengths. Besides attenuation, the signal intensity of P2T also depends on the fluence intensity of the imaging beams. For quantitative imaging, the P2T images are normalized by the fluence to correct for bias due to variation in the excitation X-ray beam fluence. Details on the fluence and attenuation correction methods can be found in Methods.

A summary of the correction methods, imaging approaches and the assumed detector time resolution used for all P2T images can be found in Table 1.

## P2T linearity with high-$Z$ elements

Both the CT and P2T image intensities are determined by the cross-section of physical interactions, or the attenuation coefficient, which is a function of the atomic number $Z$, the density $\rho$ and the photon energy hv.

In P2T with MV X-ray as the source, both Compton scatter and pair production contribute to the interaction. However, since the P2T detectors remove the majority of Compton scatter photons, the P2T image intensity overwhelmingly depends on the probability of pair-production interaction, which is linearly proportional to $\rho Z$. Consequently, the P2T image contrast should follow a simple linear relationship with $\rho Z$. If $\rho$ is known, then the atomic number $Z$ is determined. Details on the linear relationship can be found in Methods.

In comparison, the CT image signals using a kV source are produced by a mixture of Rayleigh scatter, photoelectric effect and Compton scatter, the mixture being both material and energy dependent. The Compton attenuation coefficient is approximately $Z$ independent, and the photoelectric effect is approximately proportional to $Z^3$ with sharp discontinuities at the $K$-edges, which give CT excellent sensitivity to materials with mid to high atomic numbers. On the other hand, the convolution of poly-energetic X-rays with nonlinear cross-intersections inevitably renders multiple material differentiation tasks underdetermined.

The linearity of P2T contrast to $Z$ were evaluated on an elliptical water-equivalent phantom with 10 inserts, among which 7 inserts were made up of water and 5% of high-$Z$ elements (ranging from 53 to 83), including iodine, barium, gadolinium, ytterbium, tantalum, gold and bismuth. The minor and major axes of the phantom were 20 cm and 24 cm, respectively. P2T MC simulation utilized a total of 56.6 billion primary particles in 20 equally distributed co-planar fan beams, with full coverage of the phantom in each beam. For SPB, the pencil beam size was $0.2 \times 0.2$ cm$^2$. The pencil beams were excited sequentially, and together all pencil beams covered the entire phantom at each of the 20 beam angles. MC CT simulation utilized a total of 72 billion primary particles in 360 equally distributed co-planar fan beams. The CT detector

pixel size was 0.2 cm by 0.2 cm, the source to detector distance was 100 cm and the source to isocentre distance was 66.7 cm. The beam energies of P2T and CT were 10 MV and 120 kVp, respectively. X-rays of 10 MV have a typical poly-energetic bremsstrahlung X-ray spectrum for radiotherapy. The 120 kVp X-ray spectrum is typical of a diagnostic hot cathode system. The reconstructed image resolution was 0.2 cm.

The CT and P2T images are presented in panels a and c in Fig. 2, respectively. Linear regressions of the increased contrast to water on the atomic number $Z$ are presented in Fig. 2b. The CT has a higher contrast for the high-$Z$ materials due to the $Z^3$ photoelectric cross-section, but the relationship between CT image intensity and the atomic number is nonlinear. For example, although gadolinium has a lower atomic number than ytterbium, tantalum, gold and bismuth, its $K$-edge energy at 50 keV is closer to the peak of the 120 kVp CT spectrum. Consequently, the CT contrast of gadolinium is substantially higher than those of the other materials. In comparison, the P2T image intensities show the expected linear relationship with the atomic number. The $r^2$ values of CT, P2T GT, P2T FBP, P2T SPB and P2T TOF images are 0.23, 0.99, 0.48, 0.93 and 0.84, respectively. Apart from P2T FBP, where the image contrast is obscured by excessive noise, all other P2T images show a strong linear relationship with $Z$.

## P2T linearity with tissue equivalent materials

Besides high-$Z$ nanoparticle imaging, P2T for human tissue imaging is evaluated on the same elliptical phantom containing 10 different tissue-mimicking inserts, including air, lung inhale, lung exhale, adipose, breast, water, muscle, liver, trabecular bone and dense bone (Fig. 3a), under the same geometry and energy setup. Figure 3b shows the image contrast of the P2T from different reconstruction methods compared with CT on the standard phantom, with error bars showing the standard deviations. The dashed lines show the theoretical values of P2T contrast, defined as the increments in $\rho Z_{\text{eff}}$ of each material relative to water, where $\rho$ is the material mass density and $Z_{\text{eff}}$ is the effective atomic number of a composite material[34] (Supplementary Table 1). The ground-truth and reconstructed P2T images and the CT image are shown in Fig. 3c.

Among the three P2T reconstruction methods, FBP provides the lowest SNR, making it more difficult to discern materials with $\rho Z_{\text{eff}}$ similar to water. The SPB image is comparable to the TOF image without requiring a high detector time resolution. The SPB and TOF images are noisier than the ground-truth image due to the low detector coverage of the solid angles.

For low-$Z$ materials, the photoelectric component in CT is negligible, and the contrast is approximately linear to $\rho$, while P2T is linear to $\rho Z_{\text{eff}}$. The $Z_{\text{eff}}$ factor offers greater contrast (the corresponding rods indicated by the red arrows in Fig. 3c are more visible in P2T images) for materials including the lung inhale, lung exhale, adipose and breast tissue. Specifically for the breast tissue with a 1% difference in density to water but a 13.6% difference in $Z_{\text{eff}}$, this translates to a 13.6× increase in the contrast.

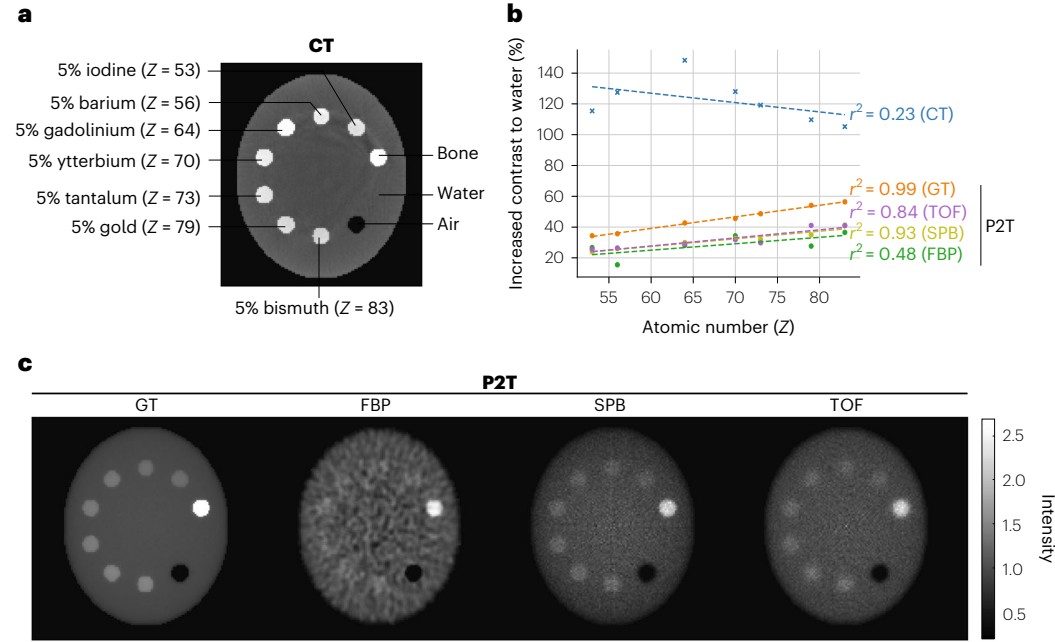

**Fig. 2 | Phantom study on P2T linearity to atomic number. a**, The CT image of a nanoparticle phantom with 10 inserts, among which 7 inserts were made up of water and 5% of high-$Z$ elements, including iodine, barium, gadolinium, ytterbium, tantalum, gold and bismuth. **b**, The relative increase in contrast to water was evaluated for the 7 inserts for the CT image and all P2T images. Linear regression of the increased contrast on the atomic number $Z$ was performed for all images. **c**, Comparison of P2T GT image, P2T image from FBP reconstruction, P2T image from SPB-based reconstruction and P2T image from TOF reconstruction. All images were normalized such that the intensity of the water insert is 1. In the specific case, the maximal imaging dose was around 3.6 cGy, assuming 100% detector efficiency.

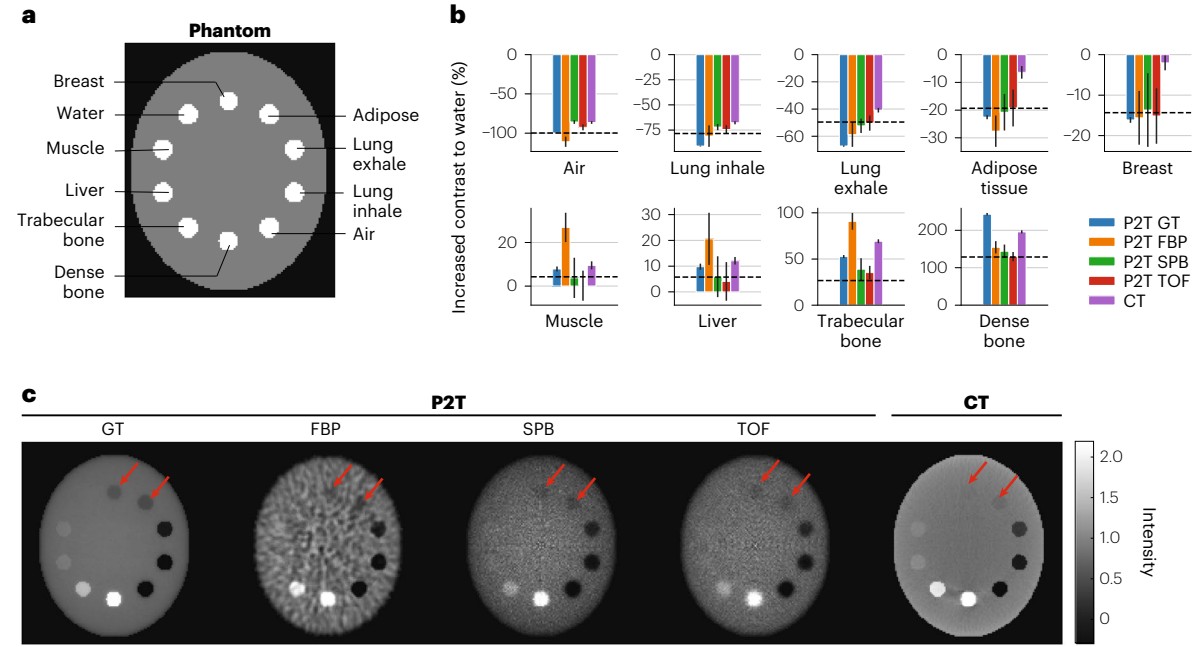

**Fig. 3 | Phantom study of standard materials. a**, The standard phantom with 10 inserts, for air, lung inhale and lung exhale, and for adipose tissue, breast tissue, water, muscle tissue, liver tissue, trabecular bone and dense bone. **b**, The relative increase in contrast to water was evaluated for all materials except water. Each bar in the bar plots is the mean over $n = 25$ independent samples. Data are presented as mean ± s.d. The dashed lines show the increments in $\rho Z_{eff}$ of each material relative to water. **c**, Comparison of P2T GT image, P2T image from FBP reconstruction, P2T image from SPB-based reconstruction, P2T image from TOF reconstruction and CT image. All images were normalized such that the intensity of the water insert is 1.

## P2T allows partial-view and sparse-view imaging

The data sufficiency condition of P2T is distinctly different from that of CT, which requires the voxel to be reconstructed on the line connecting two points on the source trajectory[35]. The requirement is translated into densely sampled full-view projections around the image subject. P2T is intrinsically compatible with partial-view and sparse-view imaging as the pair-production event detections are separable from each other. Even by locally irradiating an ROI, P2T can extract information

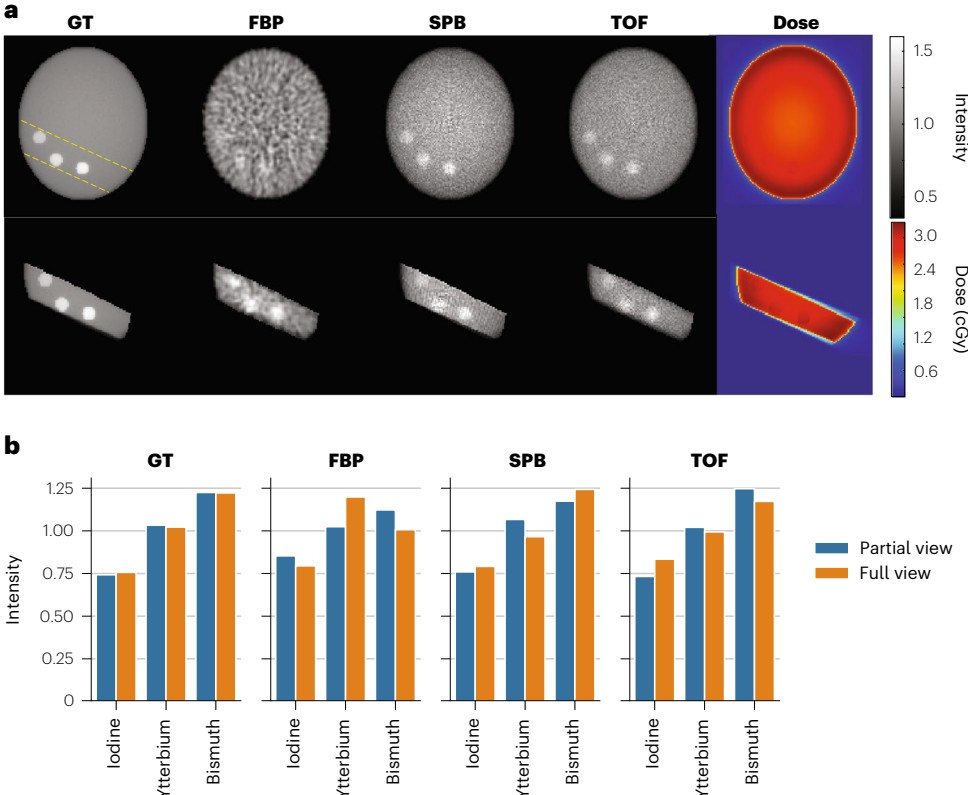

**Fig. 4 | P2T allows partial-view and sparse-view imaging. a,** Comparison of 20-beam full-view P2T images (top) and 2-beam partial-view P2T images (bottom), including GT image, P2T image from FBP reconstruction, P2T image from SPB-based reconstruction and P2T image from TOF reconstruction. All images were normalized such that the intensity of the water insert is 1. The three inserts (white spots) in the partial-view from left to right are iodine, ytterbium and bismuth. **b,** The image intensity of the 3 inserts normalized by their average values for both full-view P2T images and partial-view P2T images.

from an interior patient sub-volume. Note that the fluence correction and attenuation correction in P2T reconstruction still require the X-ray attenuation coefficients of the entire patient volume, but the corrections are insensitive to minor structures. One CT scan with the lowest possible dose could serve for fluence correction of repeated P2T acquisitions on the same imaging object.

Figure 4a shows a comparison of 20-beam full-view P2T images (top) and 2-beam partial-view P2T images (bottom). The ROIs in this case are the three inserts (5% iodine, ytterbium and bismuth from left to right) at the bottom of the phantom. The full-view P2T simulation utilized a total of 56.6 billion primary particles in 20 equally distributed co-planar beams, with full coverage of the phantom in each beam. The partial-view P2T utilized a total of 10.6 billion primary particles in 2 opposing beams, with partial beam coverage of the area indicated by the yellow dashed lines in Fig. 4a (top left). Despite irradiating only 20% of the whole volume and using only two beams, the 2-beam partial-view images are comparable to the 20-beam full-view P2T images within the ROI. More importantly, the imaging dose is limited to the irradiated volume. In the specific case, the maximal imaging dose is around 3.3 cGy, assuming 100% detector efficiency. Note that the estimated imaging dose is inversely proportional to the detector efficiency assuming fixed image SNRs. The three imaging acquisition methods (FBP, TOF and SPB) were simulated using the same number of particles and therefore have the same imaging dose.

Figure 4b shows the image intensity of the 3 inserts normalized by their average values. The ground-truth, SPB and TOF images show similar image intensity values between the 2-beam partial-view images and the 20-beam full-view images. The variations in the FBP images are due to statistical imaging noise.

## Real-time radiation-dose monitoring

Radiotherapy treatment uses high-energy X-rays (for example, 10 MV X-rays) to kill cancer cells. The same energy beams are conducive for P2T. The radiotherapy dose is closely related to the total energy released per unit mass (TERMA). TERMA is the energy loss of primary photons as they interact in the medium. It is proportional to the fluence intensity, the energy of the primary photon and the total attenuation coefficient. The radiation dose is the local energy deposition from both primary photons and secondary particles. The dose can be computed either through Monte Carlo simulation or by convolving TERMA with energy deposition kernels[36] to account for the energy spread owing to the finite travelling of secondary particles. In this study, we used Monte Carlo simulations to compute dosing. The number of pair-production interactions is proportional to the fluence intensity and the pair-production attenuation coefficient. The P2T images, defined as the number of annihilation events, show the number of pair-production interactions convoluted with a kernel representing the statistical probabilities of positron travelling before annihilation within the medium.

We tested the feasibility of obtaining P2T images using pair-production signals produced by radiotherapy beams in a patient with glioblastoma multiforme. The dose calculation was performed using Geant4 for an intensity-modulated radiotherapy (IMRT) plan with 7-equal-spacing co-planar beams. The pencil beam size for dose calculation was $0.5 \times 0.5$ cm$^2$. The dose voxel size was $0.25 \times 0.25 \times 0.25$ cm$^3$. The source is a 10 MV poly-energetic X-ray point source. The dose calculation simulated $10^8$ X-ray photons within each pencil beam. A dose matrix was constructed on the basis of the dose calculation result, which converts the x-ray fluence intensity to dose distribution within the patient body. The dose matrix was used to create a treatment plan

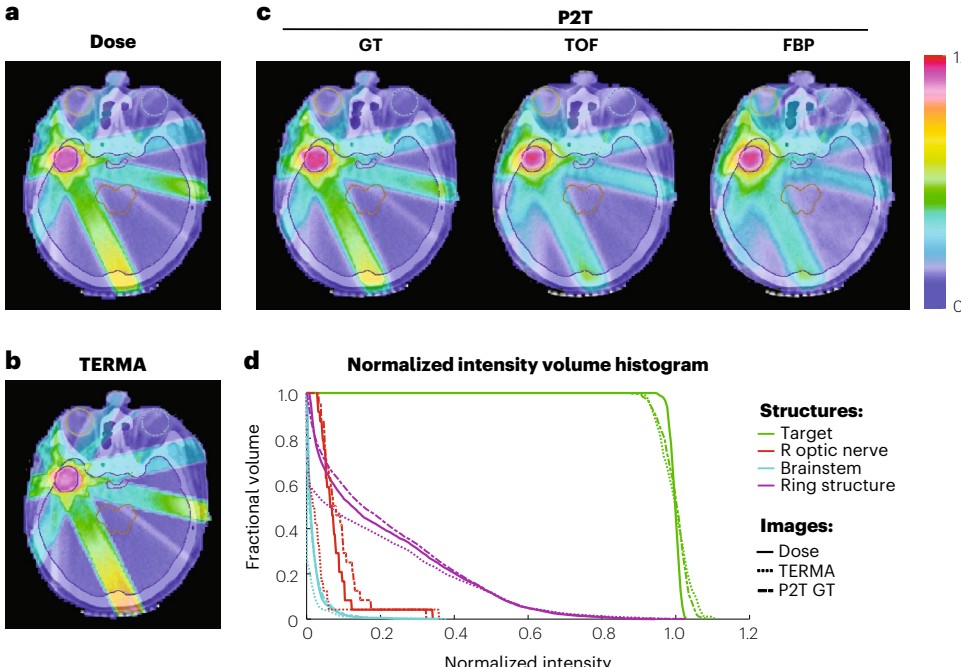

**Fig. 5 | Dose monitoring of the radiotherapy treatment for a patient with glioblastoma multiforme, using 10 MV X-ray beams and IMRT. a–c,** All dose (**a**), TERMA (**b**) and P2T (**c**) images are displayed as iso-intensity colour-wash images superimposed on the CT image. The target and the normal tissues are contoured with different colours. All colour-wash images were normalized by the mean intensity value within the target. **d,** The cumulative intensity volume histograms (cIVHs) of dose, TERMA and P2T GT images. The cIVH lines indicate the volume percentage of a structure receiving intensity values higher than a threshold.

for the IMRT delivery technique[37,38], where the radiation dose distribution was optimized to achieve prescription dose within the target volume and minimal dose to the surrounding normal tissues, using convex optimization algorithms[39,40].

The treatment plan optimization produces an optimized fluence map, dictating the number of particles in each pencil beam to achieve the optimized treatment plan. Assuming that the detector efficiency is 10% to collect a coincident photon pair, we simulated 10% of the particles needed to deliver a 2 Gy fraction treatment, which amounts to a total of 221.2 billion primary particles.

The radiotherapy treatment dose (Fig. 5a) and the TERMA (Fig. 5b) are compared with P2T images (Fig. 5c). The image resolution is $0.25 \times 0.25$ cm². All images were normalized by the mean intensity value within the target and were displayed as iso-intensity colour-wash images superimposed on the CT image. The target and normal tissues are contoured with different colours. Through optimization, the radiation dose was pushed towards the prescription dose of 2 Gy within the target and was tailored to avoid important normal tissues, including the brainstem, chiasm and eyes. Figure 5d shows the cumulative intensity volume histograms (cIVHs) of dose, TERMA and P2T ground truth. The cIVH lines indicate the volume percentage of a structure receiving intensity values greater than a threshold.

As closely linked physical quantities, the dose, TERMA and P2T are also correlated, as shown in the intensity maps and the cIVH lines. The highest intensity values are achieved within the target and the ring structure (a 1.5 cm shell surrounding the target). Sparing of normal tissues, including the brainstem, eyes, optical nerves and the majority of the brain, is verified.

In radiotherapy treatment, the number of particles and the radiation dose is 2–3 orders of magnitude greater than that for imaging, producing high SNR images even with FBP reconstruction. The differences between the ground-truth and the reconstructed images are largely due to a low detection resolution in the

patient's superior-inferior direction. The ground truth was computed with a 0.25 cm resolution in this direction, while the P2T images are effectively weighted sums of multiple image slices due to the 10 cm detector height. Note that although current radiotherapy treatment combines multiple pencil beams as an aperture for efficient delivery, SPB is a viable option with existing hardware using multileaf collimator (MLC)[41] or with a magnetic scanning beam system under development[42].

## Discussion

We report an X-ray tomography method, P2T, for radiotherapy dose verification and material imaging. Owing to the distinct image formation mechanism based on high-energy X-ray pair production, P2T provides three unique features that are distinct from the capabilities of X-ray CT. First, P2T intensities are closely related to dose for in vivo dosimetry. Unlike X-ray-induced radiation acoustic imaging[43] or Cerenkov imaging[25], in vivo dosimetry using P2T is not limited by anatomical locations and acoustic boundaries. Second, compared with CT with the photoelectric component, P2T is not as sensitive to high-$Z$ materials. Nonetheless, the P2T image intensity is linear to the atomic number after fluence correction, helping disambiguate the difficult atomic-number mapping task. The current application for multiple imaging contrast differentiation is limited by the few elements approved for clinical use, while the other elements in Fig. 2a, including tantalum, ytterbium, gold and bismuth, are still in the preclinical stage as potential imaging contrast agents. For non-contrast use, the linearity to effective atomic numbers leads to a remarkable increase in P2T contrast for certain low-atomic-number soft tissues, complementing CT's sensitivity to mid and high-$Z$ materials. Third, unlike CT that generally exposes a large volume to imaging dose from many angles, P2T can image a partial volume with as few as one beam (the sparse-view study used only two beams and 20% of the full-view), allowing geometrical

control over the imaging dose distribution. These highly generalizable features can profoundly influence a broad range of detection and diagnosis applications.

Despite the theoretical analysis of P2T in this study, its acquisition is within the realm of existing technologies. P2T benefits from decades of technological development of PET, which provides the energy and timing windows to remove non-pair-production photon contamination. Moreover, for emission-guided radiotherapy, the integration of a high-energy source and a PET detector ring was recently demonstrated[44].

We note that MV X-rays produced by linear accelerators are not readily available in a typical diagnostic department for general imaging applications. Therefore, the early development of the P2T will probably start at radiation oncology as a means for dose verification and image-guided radiation therapy. The role of P2T may later expand to the imaging realm as a complementing technology to CT, with future integration of therapy and diagnosis.

We implemented three reconstruction methods for P2T. FBP is the least technically demanding, which can be acquired with a regular PET detector and medical linac. Currently, the FBP images are only useful in high-dose therapeutic mode, and the low-SNR FBP images under low-dose P2T acquisition are only included as a reference. Given its poor image quality, iterative reconstruction methods using total variation are unlikely to improve the image quality by much. The low-dose FBP P2T may be possible if using a different detector setup with higher geometric efficiency. On the other hand, the SPB and TOF methods result in substantially higher SNRs compared with the FBP method with the same geometry setup. We point out that ultrafast TOF detectors are an active area of research, with many technical challenges to balance time resolution and detecting efficiency. In the study, we simulated detectors with a 20 ps time resolution to localize the annihilation event with an accuracy of 3 mm. The 20 ps time resolution detector is consistent with the roadmap of PET detection using prompt Cherenkov emission[31,32] or ultrafast emitting quantum-confined systems[45,46], but both still require significant engineering development to be practical[47]. On the other hand, the SPB does not require fast TOF detectors and is readily achievable using current technology. It largely relaxes the required time resolution due to the known excitation path. We assumed a 300 ps time resolution for the SPB-based and FBP reconstruction, which is commercially available for PET[33]. The sequential pencil beams required for SPB are also feasible with current technologies, such as MLC[41] or scanning photon beams[42].

## Methods

### Simulation of the detection signal

We assumed that the primary photons within each pencil beam are released from the source following a uniform distribution with an average releasing rate $R$. At the time of the incident, photon generation $t_{incident}$ is:

$$t_{incident}(n, b) = t_{incident}(n-1, b) + \tilde{t}, \, \tilde{t} \sim U(0, 2/R)$$

where $n$ is the index of the released primary photon, $b$ is the index of the pencil beam. In the volume excitation, the first primary photons of all pencil beams are released altogether. In scanning pencil beam excitation, the first primary photon of one pencil beam only starts after all photons are released in the previous pencil beam.

When a qualifying photon (within the energy resolution window) passes through the ring detector, the colliding detector module records the travelling time $t_{travel}$ of the detection event since the primary photon was generated and departed from the source, on the basis of the MC simulation in Geant4. The global time $t_{global}$ of the detection is then computed by adding the travelling time $t_{travel}$ to the generation time of the corresponding incident photon $t_{incident}$:

$$t_{global} = t_{incident} + t_{travel}.$$

The detector response time $t_{response}$ is simulated as a Gaussian distribution with variance $\sigma^2 = \Delta T^2$, where $\Delta T$ is the time resolution of the detector. The simulated detection time $t_{detection}$ of the photon is:

$$t_{detection} = t_{global} + t_{response}, \, t_{response} \sim N(0, \Delta T^2)$$

For each beam, the detected signals were discarded for all detector modules receiving primary photons due to the difficulty of identifying the annihilation photons from the primary photons with energy near 511 keV.

Some simplifications in detector geometry and properties were made in this study. For simplicity, we assumed an ideal point source without leakage. In reality, a shielding structure is required to remove X-ray photons from source leakage (~1% of primary X-ray photons), so that they would not interfere with the P2T signals. We also assumed an ideal ring detector without electronic noise and cross-talks in adjacent detector elements. In reality, the detector response would decrease the image SNR and resolution. In addition, the photon may travel through a few detector modules before generating a signal, causing parallax error. The parallax error could be avoided with a more advanced detector using depth-of-interaction information[48,49]. We also assumed an ideal geometry for CT. The poly-energetic CT source was modelled as a single point source with 4.3 mm Al filtration, and we assumed an ideal detector response with no cross-talks. The simulation was based on a very thin fan-beam geometry (5 mm fan beam) and an anti-scatter grid was not simulated. The impacts of non-ideal geometry on image quality can be found in the literature[50–53].

Despite assuming an ideal detector response, the SNRs of the reconstructed images are still lower than the ground-truth P2T images. The low SNR can be attributed to a wide detector module (a single detector module is 10 cm long in the patient longitudinal direction) and an extremely low detector geometric efficiency: the ring detector only covers 4.17% of the $4\pi$ space, leading to only $(4.17\%)^2 = 0.17\%$ efficiency for collecting coincident photon pairs. Using detectors with multiple rows and extending the longitudinal coverage to 2 m (such as in the EXPLORER project for total-body PET scanner[54]) could improve the image resolution and raise the detector geometric efficiency by 2 orders of magnitude. These improvements are expected to improve the SNR, shorten the imaging time and reduce the radiation dose.

We used the general-purpose Monte Carlo package Geant4 (ref. [29]) to study the P2T performance. Geant4 includes numerous well-validated physical models and is flexible for different applications, but as a CPU-based software package, it is extremely slow. To accelerate the simulation, we developed an automated and distributed computation framework that allows asynchronous and scalable computation divided at the unit of individual pencil beams. A total of over 280 logical CPU cores were utilized for simulation. The simulation time for the full-view phantom simulation, partial-view phantom simulation and radiotherapy imaging simulation are 30 h, 7 h and 6 d, respectively. Further acceleration may be achieved by using the graphic processing unit (GPU)-based MC code with simplified physical models[55].

### FBP reconstruction

From the simulated detector signals, coincident events were identified as two energy-eligible photons (within the energy window) arriving at two detector modules within the coincidence time $t_{coincidence}$. Two coincident events define a LOR, the line connecting the two detector modules, indicating that the annihilation event happened on the LOR.

With all LORs identified, a rebinning algorithm was applied to convert the list-mode pair-wise detector data to the sinogram data. The list-mode data of 1,440 detectors were histogrammed into sinograms having 227 radial bins and 1,440 angles. FBP reconstruction was applied to the sinogram data using the MIRT[30], where a plain ramp filter was

applied on the Fourier transform of the sinogram at each angle before applying an inverse Fourier transform and back projection.

## SPB reconstruction

SPB uses SPBE, and the locations of annihilation events can be further tracked down to the area where the incident beam passes through. The pencil beam width was 2 mm in this study. The SPB-based reconstruction was applied to the list-mode pair-wise detector data. The intersection of each detector pair's corresponding LOR and pencil beam path locates the annihilation event. The SPB reconstruction tallied the intersections within each voxel from all detector-pair data. Each intersection point was locally convolved with a Gaussian kernel ($\sigma$ = 2 mm in this study) to reduce image noises and artefacts.

## TOF reconstruction

For detectors with a high time resolution (such as 20 ps), the location of the annihilation event can be computed from the flying time of the two photons. The travelling distance difference $\Delta d$ between the two photons is $\Delta d = c\Delta t$, where $c$ is the speed of light and $\Delta t$ is the time difference of the two photons when arriving at the detectors. The location of the annihilation event on the LOR can then be derived from the travelling distance difference $\Delta d$. The located point was locally convolved with a Gaussian kernel ($\sigma$ = 2 mm in this study) to reduce image noises and artefacts. Note that the image resolution $\Delta R = c\Delta T/2$ is limited by the detector time resolution $\Delta T$.

## Attenuation correction

Before arriving at the detectors, the two coincident photons may be absorbed or scattered as they travel through the imaging subject. To compensate for the attenuation, the detector data need to be corrected accordingly.

For FBP, the attenuation correction was performed on the sinogram:

$$P_c(i) = P_r(i) \exp\left(\int_i \mu_{511}\, dl\right),$$

where $P_c$ is the corrected sinogram and $P_r$ is the raw sinogram obtained directly from the list-mode detector data. $i$ is the index of the sinogram. $\mu_{511}$ is the attenuation coefficient of the imaging subject for 511 keV X-ray. $dl$ stands for differential of the variable $l$, which is the path length of the X-ray photon. The attenuation correction factor $\exp(\int_i \mu_{511}\, dl)$ is an integral over the path of the coincident photons associated with the $i$th element of the sinogram. After the attenuation correction, the corrected sinogram $P_c$ was used for FBP reconstruction.

For the SPB-based method and the TOF method, the attenuation correction factor was computed as the same line integral over the path of the coincident photons. During reconstruction, the voxel-wise tallies were weighted by the correction factor of each identified detector pair.

## Fluence correction for quantitative imaging

The total nuclear pair-production cross-section per atom $\alpha_a$ is[28]:

$$\alpha_a = \sigma_0 Z^2 \int_0^1 P d\left(\frac{T^+}{hv - 2m_0 c^2}\right) = \sigma_0 Z^2 \bar{P} \,\tilde{\propto}\, Z^2$$

where $T^+$ is the positron energy, $hv$ is the energy of the incident photon, $Z$ is the atomic number, $P$ is a function of $hv$ and $Z$, $m_0$ is the mass of electron and positron, $\sigma_0$ is a constant.

The attenuation coefficient $\alpha$ of nuclear pair production is therefore proportional to the atomic number $Z$ and density $\rho$:

$$\alpha = \rho \alpha_a \frac{N_0}{A} \propto \rho Z,$$

where $A$ is the atomic mass number and $N_0$ is the Avogadro constant. A simplification was made using the property that $Z/A$ is close to 1 for most elements.

The P2T image signal is proportional to the attenuation coefficient of the imaging material and the X-ray fluence intensity $f$:

$$I \propto f\alpha \propto f\rho Z$$

The fluence intensity at image voxel $v$ from pencil beam $b$ is

$$f_{v,b} = f_{r,b} \frac{(p_r - p_s)^2}{(p_v - p_s)^2} \exp\left(-\int_{p_s}^{p_v} \mu_{10MV}\, dl\right),$$

where $p_s$, $p_v$ and $p_r$ are the locations of the source, image voxel $v$ and reference point $r$, respectively. The reference point $r$ is outside the imaging subject and on the line segment connecting the source and the voxel $v$. $f_{r,b}$ is the fluence intensity at the reference point from pencil beam $b$, and $f_{v,b}$ is the fluence intensity at image voxel $v$ from pencil beam $b$. $\mu_{10MV}$ is the attenuation coefficient of the imaging subject for 10 MV X-ray. The integral was computed using Siddon's ray tracing algorithm[56] with the matRad toolbox[57].

The total fluence intensity at image voxel $v$ is

$$f_v = \sum_b f_{v,b}.$$

The dependence of the image intensity on the incident X-ray fluence intensity can be removed with the fluence correction. The corrected image intensity $\bar{I}$ is proportional to the product of density and the atomic number, obtained by voxel-wise scaling of the P2T image by the fluence intensity $f$:

$$\bar{I}_v = I_v / f_v \propto \rho_v Z_v.$$

Note that the fluence correction was only performed for quantitative imaging applications and did not apply to the P2T images obtained during radiotherapy treatments.

## P2T image contrast for compounds

The Bragg's additivity rule applies to the pair-production mass attenuation coefficient of compounds $\left(\frac{\alpha}{\rho}\right)_{comp}$:

$$\left(\frac{\alpha}{\rho}\right)_{comp} = \sum_i \left(\frac{\alpha}{\rho}\right)_i f_i$$

where $f_i$ and $\left(\frac{\alpha}{\rho}\right)_i$ are the weight fraction and the pair-production mass attenuation coefficient of element $i$, respectively. After fluence correction, the P2T image intensity $\bar{I}$ is proportional to the pair-production attenuation coefficient of compounds $(\alpha)_{comp}$. Using the property that $\left(\frac{\alpha}{\rho}\right)_i \propto Z_i$, then

$$\bar{I} \propto (\alpha)_{comp} \propto (\rho)_{comp} \sum_i Z_i f_i,$$

where $Z_i$ is the atomic number of element $i$, and $(\rho)_{comp}$ is the density of the compounds. Let $Z_{eff}$ be the effective atomic number

$$Z_{eff} = \sum_i Z_i f_i,$$

then the P2T image intensity $\tilde{I}$ is proportional to $(\rho)_\text{comp} Z_\text{eff}$:

$$\tilde{I} \propto (\rho)_\text{comp} Z_\text{eff}.$$

Therefore, $(\rho)_\text{comp} Z_\text{eff}$ provides a theoretical value of P2T image contrast.

The material composition[58], density and effective atomic number of the 10 inserts in the standard phantom are listed in Supplementary Table 1.

### Radiotherapy dose and P2T

The radiotherapy dose is closely related to a concept named TERMA, which is defined for photons as

$$\text{TERMA} = \int f(E) E \frac{\mu(E)}{\rho} dE$$

where $f$ is the fluence intensity, $E$ is the photon energy, $\rho$ is the material density and $\mu$ is the attenuation coefficient, including contributions from Rayleigh, photoelectric, Compton and pair-production interactions.

Radiation dose can be computed through direct Monte Carlo simulation or through an analytical approach, where TERMA is convoluted with energy deposition kernels using the collapsed cone convolution algorithm[36]. These Monte-Carlo precomputed energy deposition kernels account for the energy spread due to finite travelling of secondary particles, with different dose spreads under different materials and photon energies.

The P2T image collected during radiotherapy is also proportional to the fluence intensity (no fluence correction applied):

$$I \propto \int f(E) \alpha(E) dE$$

where $\alpha(E)$ is the pair-production attenuation coefficient.

Most human tissues are approximately water-equivalent under the high-energy X-rays used in radiotherapy. Therefore, the P2T image intensity and the TERMA are proportional to the fluence intensity. Consequently, the P2T image is strongly correlated with the dose distribution and can be used for in vivo dose monitoring.

### Positron travelling before annihilation

Supplementary Fig. 1 shows histograms of the positron travelling distances before annihilation and the initial positron energies from pair production. The median of the positron travelling distances before annihilation was 4.6 mm, and the median of the initial positron kinetic energy was 1.1 MeV. The finite positron travels contribute to the theoretical image resolution of raw P2T images. On the other hand, the positron travelling distances histogram is determined by the incident photon energy spectrum. Super-resolution P2T images can be recovered through deconvolution with the precomputed kernels.

### Reporting summary

Further information on research design is available in the Nature Research Reporting Summary linked to this article.

## Data availability

The main data supporting the results in this study are available within the paper and its Supplementary Information. All phantom data used to generate simulation data in this study are available on Zenodo with the identifier https://doi.org/10.5281/zenodo.6330603. The raw and analysed P2T detector data generated during the study are too large to be publicly shared, yet they are available for research purposes from the corresponding author on reasonable request.

## Code availability

All codes used for data acquisition and for data analysis are available on Zenodo with the identifier https://doi.org/10.5281/zenodo.6330603.

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

## Acknowledgements

This research is supported by DOE Grant Nos. DE-SC0017057 (K.S.) and DE-SC0017687 (K.S.), NIH Grant Nos. R01CA188300 (K.S.), R43CA183390 (K.S.) and R44CA183390 (K.S.).

## Author contributions

All authors contributed extensively to the study. K.S. conceived the study and designed the experiments. Q.L. designed and implemented Monte Carlo modelling and image reconstruction, and analysed the results for P2T and CT. R.N. implemented Monte Carlo modelling for dose calculations and the distributed CPU computing framework for acceleration. All authors discussed the results and implications, and commented on the manuscript.

## Competing interests

The authors declare no competing interests.

## Additional information

**Correspondence and requests for materials** should be addressed to Ke Sheng.

# Reporting Summary

## Statistics

For all statistical analyses, confirm that the following items are present in the figure legend, table legend, main text, or Methods section.

| n/a | Confirmed | |
|---|---|---|
| ☒ | ☐ | The exact sample size (*n*) for each experimental group/condition, given as a discrete number and unit of measurement |
| ☒ | ☐ | A statement on whether measurements were taken from distinct samples or whether the same sample was measured repeatedly |
| ☒ | ☐ | The statistical test(s) used AND whether they are one- or two-sided<br>*Only common tests should be described solely by name; describe more complex techniques in the Methods section.* |
| ☒ | ☐ | A description of all covariates tested |
| ☒ | ☐ | A description of any assumptions or corrections, such as tests of normality and adjustment for multiple comparisons |
| ☒ | ☐ | A full description of the statistical parameters including central tendency (e.g. means) or other basic estimates (e.g. regression coefficient) AND variation (e.g. standard deviation) or associated estimates of uncertainty (e.g. confidence intervals) |
| ☒ | ☐ | For null hypothesis testing, the test statistic (e.g. $F$, $t$, $r$) with confidence intervals, effect sizes, degrees of freedom and $P$ value noted<br>*Give P values as exact values whenever suitable.* |
| ☒ | ☐ | For Bayesian analysis, information on the choice of priors and Markov chain Monte Carlo settings |
| ☒ | ☐ | For hierarchical and complex designs, identification of the appropriate level for tests and full reporting of outcomes |
| ☒ | ☐ | Estimates of effect sizes (e.g. Cohen's *d*, Pearson's *r*), indicating how they were calculated |

*Our web collection on statistics for biologists contains articles on many of the points above.*

## Software and code

Policy information about availability of computer code

| Data collection | All codes used for data acquisition are available on Zenodo with the identifier https://doi.org/10.5281/zenodo.6330603. |
|---|---|
| Data analysis | All codes used for data analysis are available on Zenodo with the identifier https://doi.org/10.5281/zenodo.6330603. |

For manuscripts utilizing custom algorithms or software that are central to the research but not yet described in published literature, software must be made available to editors and reviewers. We strongly encourage code deposition in a community repository (e.g. GitHub). See the Nature Portfolio guidelines for submitting code & software for further information.

## Data

Policy information about availability of data

All manuscripts must include a data availability statement. This statement should provide the following information, where applicable:
- Accession codes, unique identifiers, or web links for publicly available datasets
- A description of any restrictions on data availability
- For clinical datasets or third party data, please ensure that the statement adheres to our policy

The main data supporting the results in this study are available within the paper and its Supplementary Information. All phantom data used to generate simulation data in this study are available on Zenodo with the identifier https://doi.org/10.5281/zenodo.6330603. The raw and analysed P2T detector data generated during the study are too large to be publicly shared, yet they are available for research purposes from the corresponding author on reasonable request.

# Field-specific reporting

Please select the one below that is the best fit for your research. If you are not sure, read the appropriate sections before making your selection.

☒ Life sciences ☐ Behavioural & social sciences ☐ Ecological, evolutionary & environmental sciences

For a reference copy of the document with all sections, see nature.com/documents/nr-reporting-summary-flat.pdf

# Life sciences study design

All studies must disclose on these points even when the disclosure is negative.

| | |
|---|---|
| Sample size | Not applicable, as the study used computer simulations based on first principles. |
| Data exclusions | – |
| Replication | – |
| Randomization | – |
| Blinding | – |

# Reporting for specific materials, systems and methods

We require information from authors about some types of materials, experimental systems and methods used in many studies. Here, indicate whether each material, system or method listed is relevant to your study. If you are not sure if a list item applies to your research, read the appropriate section before selecting a response.

## Materials & experimental systems

| n/a | Involved in the study |
|---|---|
| ☒ ☐ | Antibodies |
| ☒ ☐ | Eukaryotic cell lines |
| ☒ ☐ | Palaeontology and archaeology |
| ☒ ☐ | Animals and other organisms |
| ☒ ☐ | Human research participants |
| ☒ ☐ | Clinical data |
| ☒ ☐ | Dual use research of concern |

## Methods

| n/a | Involved in the study |
|---|---|
| ☒ ☐ | ChIP-seq |
| ☒ ☐ | Flow cytometry |
| ☒ ☐ | MRI-based neuroimaging |

