## [Peer Review File · Nature Biomedical Engineering]

Tomographic detection of photon pairs produced from high-energy X-rays for the monitoring of radiotherapy dosing

Corresponding author: Ke Sheng

Editorial note

This document includes relevant written communications between the manuscript's corresponding author and the editor and reviewers of the manuscript during peer review. It includes decision letters relaying any editorial points and peer-review reports, and the authors' replies to these (under 'Rebuttal' headings). The editorial decisions are signed by the manuscript's handling editor, yet the editorial team and ultimately the journal's Chief Editor share responsibility for all decisions.

Any relevant documents attached to the decision letters are referred to as **Appendix #**, and can be found appended to this document. Any information deemed confidential has been redacted or removed. Earlier versions of the manuscript are not published, yet the originally submitted version may be available as a preprint. Because of editorial edits and changes during peer review, the published title of the paper and the title mentioned in below correspondence may differ.

Correspondence

Mon 19 Apr 2021

Decision on Article nBME-21-0210-T

Dear Dr Sheng,

Thank you again for submitting to *Nature Biomedical Engineering* your manuscript, "Pair production tomography imaging". The manuscript has been seen by 3 experts, whose reports you will find at the end of this message. You will see that the reviewers have good words for the work, and that they raise a number of technical criticisms that we hope you will be able to address. In particular, we would expect that a revised version of the manuscript provides:

* improved characterization of the pair production tomography feasibility, also compared to PET technologies, including signal to background ratios, scan times etc.,

* improved reconstructed image data quality,

* clarifications that address the limitations posed by the use of Monte Carlo simulations, and, if possible, a demonstration of pair production data acquired from an in vivo setting.

When you are ready to resubmit your manuscript, please upload the revised files, a point-by-point rebuttal to the comments from all reviewers, the (revised, if needed) reporting summary, and a cover letter that explains the main improvements included in the revision and responds to any points highlighted in this decision.

Please follow the following recommendations:

* Clearly highlight any amendments to the text and figures to help the reviewers and editors find and understand the changes (yet keep in mind that excessive marking can hinder readability).- * If you and your co-authors disagree with a criticism, provide the arguments to the reviewer (optionally, indicate the relevant points in the cover letter).
- * If a criticism or suggestion is not addressed, please indicate so in the rebuttal to the reviewer comments and explain the reason(s).
- * Consider including responses to any criticisms raised by more than one reviewer at the beginning of the rebuttal, in a section addressed to all reviewers.
- * The rebuttal should include the reviewer comments in point-by-point format (please note that we provide all reviewers will the reports as they appear at the end of this message).
- * Provide the rebuttal to the reviewer comments and the cover letter as separate files.

We hope that you will be able to resubmit the manuscript within 25 weeks from the receipt of this message. If this is the case, you will be protected against potential scooping. Otherwise, we will be happy to consider a revised manuscript as long as the significance of the work is not compromised by work published elsewhere or accepted for publication at *Nature Biomedical Engineering*. Because of the COVID-19 pandemic, should you be unable to carry out experimental work in the near future we advise that you reply to this message with a revision plan in the form of a preliminary point-by-point rebuttal to the comments from all reviewers that also includes a response to any points highlighted in this decision. We should then be able to provide you with additional feedback.

We hope that you will find the referee reports helpful when revising the work, which we look forward to receive. Please do not hesitate to contact me should you have any questions.

Best wishes,

Rosy

Dr Rosy Favicchio
Senior Editor, Nature Biomedical Engineering

Reviewer #1 (Report for the authors (Required)):

The authors describe a very interesting and novel tomography imaging method that utilizes the density of megavoltage X-ray beam-induced pair production events to generate true atomic density distribution maps. The study is purely theoretical, however it addresses many practical questions by approximating the limitations of real detectors and doses. The authors demonstrate three different reconstruction approaches, which cover different applications and detector types (Time-of-flight, pencil beam scanning, etc.), as well as three different real-world scenario for high-Z CT, tissue CT, and in vivo beam dosimetry.

The manuscript is exceptionally well written, clear, and addresses reader's questions seamlessly. This work has the potential to be very influential, provided that the approximations here are representative. I am not 100% convinced the signal to noise will be satisfactory with current setups and doses usual for standard CT, but it will be exciting to see this application unveil its benefits as the technology advances. The theoretical explanation is simple yet clean and understandable, and I did not find any issue with it.

I am enthusiastic about this work and it's merit for Nature Biomedical Engineering, and have only a few methodology questions and several minor comments about the figures.

CT requires much simpler, charge-integrating detector in comparison to PET. What is the required dose rate (and therefore real scan time) necessary to produce the demonstrated figures, given a certain non-zero dead time of photon counting PET detectors?

In order for energy discriminator to work efficiently, the photon counting rate shall not exceed say 1-10% of theoretical counting rate to prevent event pile-up. Considering the strong, omnidirectional scatter present due to Compton and other mechanisms, I can imagine the detectors may be flooded with this background signal, preventing the energy filtering and coincidence detection to work. Can the authors shed light on the discrimination of pair production photons and all the background? This could be included as a figure in a form of time-resolved histogram, similar to that in Fig 1, but with a representative PET detector and selected object under scan (for example the one used later in manuscript). This can definitely grow out of the scope of this manuscript very fast, but I believe the authors should take this into consideration and demonstrate it, as it seems to be pivotal for the success of this method.

Fig. 1 c)-e) is missing axis labels - even though it is explained in the caption, it would add clarity to annotate them in figures.

Likewise, the total dose used to form Figures 2 and 3 could be mentioned in caption, so it is immediately clear.

The color bars are occasionally missing a label, and the text size may be too low in final format.

I will gladly recommend the manuscript for publication once the technical comments and stylistic issues are addressed.

Reviewer #2 (Report for the authors (Required)):

See Appendix 1.

Reviewer #3 (Report for the authors (Required)):

This is a very interesting paper that introduces the concept of an X-ray stimulated emission tomography, where the incoming X-rays ($>1.022\text{MeV}$) stimulate electron/positron pair-production which in turn produces photon-pairs when the positron annihilates with an electron in the medium.

This pair-production tomography (PPT) seems very similar to PET but with a different (and more controllable) source of positrons. However, PPT which is an emission tomography seems to be constantly compared with CT, a transmission tomography method, throughout the paper.

PPT is explored using Monte-Carlo simulation through the Geant4 software. Three different image formation processes (each with different pros/cons) are compared and evaluated.

The main question that arises would appear to be: "wouldn't that method have a very large and high-energy dose on the patient?" This does not really seem to have been addressed adequately or explicitly.

A very compelling case that nullifies this question is the ability to image the dose when performing radiation therapy. This idea alone is sufficient for a great paper.

You could consider shortening the title to "Pair-production tomography" or add a bit more description: "Pair-production emission tomography"?

Have you considered doing transmission CT from the attenuation of the primary beam at the same time?

While I think this is some really great work, I'm not sure that the ideas are sufficiently developed for nature:BME. This is foundational work, but I think perhaps the first physical demonstration of PPT would more merit publication in nBME. Physical realisation of PPT would require many problems/limitations to be overcome that are completely missed by Monte-Carlo simulation.

Some specific major comments:

line 11-12: "We studied three P2T acquisition methods, including filtered back projection (FBP), time-of-flight (TOF), and scanning pencil beam (SBP)."

Filtered backprojection is not an acquisition scheme, it is a reconstruction method.

The methods should perhaps be described as something like:

"We studied three P2T acquisition methods: a scanning pencil beam (SBP) and two full-field illumination methods, one with temporal resolution sufficient to identify photon-pairs and one with higher temporal resolution sufficient for photon time-of-flight (TOF) estimation."

line 15-16:"...the ability to form tomography with as few as a single heavily truncated X-ray beam,"This sentence does not really make sense without reading the whole paper; even then, heavily truncated is a vague term - what is truncated? Perhaps you could say something like:"PPT is a form of emission tomography, so even a single direction of illuminating X-rays are sufficient for tomography and the beam can illuminate a region-of-interest (ROI) only."

line 25-26: What makes PET expensive? Wouldn't PPT be similar in expense? (unless it is the production of tracers that's the expensive part I guess?)

line 38-39: "As a result, it is difficult to obtain clean material information from the X-ray CT images."Should possibly mention dual-energy CT (DECT) as a solution for this?

line 39-41: Another weakness of CT is poor soft-tissue contrast due to the similarity in densities and the diminishing photoelectric effect with low atomic number materials. "Should possibly mention phase-contrast methods as a solution for this?"

lines 42-49: talk about region-of-interest problem in CT. This is only relevant for transmission tomography; PPT is an emission tomographic method, like PET and SPECT.

Perhaps should talk about invention/developments and applications of medical emission X-ray CT here as well, i.e., PET and SPECT, which seems more relevant to PPT.

line 50-52: "The ionizing radiation from high-energy X-rays can break DNA strands, which, if not repaired, leads to cell death."This seems serious, why would I choose to do PPT then, unless measuring dose from radiotherapy? This needs to be addressed later if PPT is being proposed outside of a dose measurement application.

line 84-85: "...the positron loses its kinetic energy and comes to a near stop, it comes into contact with an electron with nearly simultaneous annihilation..."What is a typical distance that the positron travels before annihilation? I assume that it's a very small distance in water... however, this would be a lower limit to PPT resolution right?

lines 124-134: Describes the tomographic image generation process for the three methods. This is not quite clear, needs a bit of expansion (particularly the TOF method). It seems that the first method that uses FBP is a conventional CT-type method while the TOF and SPB methods are more direct imaging techniques, i.e., not inverting Radon transform, rather just adding data to the image as it is measured. Why did you not add some regularisation to the FBP method, e.g., TV-minimisation?

line 131-133: "For detectors with a high time resolution (e.g., 20 ps), the range of the annihilation event along the LOR can be narrowed down according to the time-of-flight (TOF) of the two photons."20 picosecond resolution means that pair-production position can be determined to about $0.3 \times 10^{12} \text{ mm/s} \times 20 \times 10^{-12} \text{ s} = 6 \text{ mm}$ resolution. So this is the lower limit to resolution here? Should this be mentioned? To improve resolution, could this TOF data be reconstructed using the FBP method without timestamps as well as using the TOF information in a statistical sense?

lines 135-139: Corrections for fluence variation outlined. Where do the "radiological pathlengths" come from? are they pre-measured by CT? does this mean that true ROI is not really possible? Mention that this is explained in detail later in the paper (lines 417-439)

Table 1: Would be good to spell out acronyms in the caption. It took me a while to work out that GT was ground truth.

line 154-155: "Therefore, in theory, P2T image contrast also follows a simple linear relationship with ρZ ."

Mention that this is explained in detail later in the paper (lines 440-456)

line 159-162: "The convolution of poly-energetic X-rays with non-linear cross-intersections inevitably renders multiple material differentiation tasks an underdetermined problem." Mention that it is common to take additional information using a second X-ray spectrum (i.e., dual-energy CT) for this reason. If PPT is related to ρ and Z , isn't this also underdetermined? can't distinguish between something low density with high Z from something with high density and low Z ?

The analysis graphs presented with the images in Figs 2-4 are different every time. This makes it difficult to compare/contrast. Please make them all consistent (preferably the same as in Fig. 2 using a single chart)

line 208-209: "The ρZ factor offers greater contrast for materials including the lung inhale, lung exhale, adipose, and breast tissue." I find it hard to tell for some of these regions. Can you point the reader to specific things to look at to observe this?

lines 220-225: Again, this is comparing an emission CT (PPT) with transmission CT - not really worth doing? Maybe should instead compare PET and SPECT; similar but PPT can be in a targeted region --- could compare radiation doses for PET/SPECT/PPT?

lines 226-245: did these simulations use fluence correction? Measured using CT? this should be mentioned...

line 234-235: "In the specific case, the maximal imaging dose is around 3.3 cGy, assuming 100% detector efficiency."

Please explain how detector efficiency is related to imaging dose?

line 254: please explain what "energy deposition kernels" are and how they are determined.

line 256-258: "The P2T images, defined as the number of annihilation events, show the number of pair production interactions convoluted with the positron traveling."

Two questions:

- 1) Does the "energy" of each pair-production event need to be accumulated rather than the "count" to become more similar to TERMA? or is it sufficient just to scale by 1.022MeV per count?
- 2) Should this be something like "convolved with a kernel representing the statistical probabilities of positron interaction within the medium"

line 267: Please explain what "conformal dose" is?

line 270-272: "Assuming the detector efficiency is 10% to collect a coincident photon pair, we simulated 10% of the particles needed to deliver a 2Gy fraction treatment, which amounts to a total of 221.2 billion primary particles." I don't understand this paragraph; why do you need to assume a detector efficiency? can't you model a detector properly? If the detector is 10% efficient, wouldn't you need to deliver 10x the dose, not 10% of the dose?

Some specific minor comments:

line 23: "There is a long history of using X-rays for detection." Detection of what? probably need a reference or multiple references here...

line 27-28: "A major breakthrough in X-ray imaging is the invention of tomographic images." Probably need to reference Hounsfield or Cormack here...

line 28: "By acquiring the 1D linear or 2D planar projection images..."

This probably needs a bit more explanation, what are projection images? how can I have a 1D linear image?

line 32-33: "Since CT's invention, many technological developments in the hardware and

reconstruction methods have significantly improved its speed, image quality, and versatility."
Probably need a reference here

line 97-99: "The 511 keV photons consist 0.91%, 15.4%, and 77.6% of the total photons, before applying filters (Figure 1c), after applying an $\pm 10\%$ energy window filter (Figure 1d), and after applying the $\pm 10\%$ energy and 1 ns coincidence time filters (Figure 1e), respectively."

This seems like a confusing way to present the info; "respectively" only applies when two items are listed. Perhaps just say: "The 511 keV photons consist 0.91% of the total photons, before applying filters (Figure 1c), 15.4% after applying a $\pm 10\%$ energy window filter (Figure 1d), and 77.6% after applying the $\pm 10\%$ energy as well as a 1 ns coincidence time filters (Figure 1e)."

line 162: missing a "with" before 10

line 177-180: "For example, although the Gadolinium has a lower atomic number than Ytterbium, Tantalum, Gold, and Bismuth, its K-edge energy at 50 keV is closer to the peak of the 120 kVp CT spectrum. Consequently, the CT contrast of Gadolinium is substantially higher than other materials."
Excellent description!

line 263: "The dose calculation simulated 10 particles within each pencil beam." Should "particles" be "X-ray photons"?

Wed 28 Jul 2021

Decision on Article nBME-21-0210A

Dear Dr Sheng,

Thank you for your revised manuscript, "Pair production tomography", which has been seen by the original reviewers, and please excuse the unusual delay in reaching a decision, due to a reviewer requesting an extension. In their reports, which you will find at the end of this message, you will see that the reviewers acknowledge the improvements to the work and raise a few additional technical criticisms that we hope you will be able to address. In particular, we would expect that the next version of the manuscript provides:

* improved description of the technology proposed and of its most promising applications

As before, when you are ready to resubmit your manuscript, please upload the revised files, a point-by-point rebuttal to the comments from all reviewers, the reporting summary, and a cover letter that explains the main improvements included in the revision and responds to any points highlighted in this decision.

As a reminder, please follow the following recommendations:

* Clearly highlight any amendments to the text and figures to help the reviewers and editors find and understand the changes (yet keep in mind that excessive marking can hinder readability).

* If you and your co-authors disagree with a criticism, provide the arguments to the reviewer (optionally, indicate the relevant points in the cover letter).

* If a criticism or suggestion is not addressed, please indicate so in the rebuttal to the reviewer comments and explain the reason(s).

* Consider including responses to any criticisms raised by more than one reviewer at the beginning of the rebuttal, in a section addressed to all reviewers.

* The rebuttal should include the reviewer comments in point-by-point format (please note that we provide all reviewers will the reports as they appear at the end of this message).

* Provide the rebuttal to the reviewer comments and the cover letter as separate files.

We hope that you will be able to resubmit the manuscript within 12 weeks from the receipt of this message. If this is the case, you will be protected against potential scooping. Otherwise, we will be happy to consider a revised manuscript as long as the significance of the work is not compromised by work published elsewhere or accepted for publication at *Nature Biomedical Engineering*.

We look forward to receive a further revised version of the work. Please do not hesitate to contact me should you have any questions.

Best wishes,

Rosy

Dr Rosy Favicchio
Senior Editor, Nature Biomedical Engineering

Reviewer #1 (Report for the authors (Required)):

The authors addressed all my comments and concerns, and I'm happy to recommend the manuscript for publication. However, I still do have a few last comments that should be addressed prior the publication.

In the very first intro sentence, my suggestion would be use "imaging" instead of "detection". The latter is too broad and it really does not make too much sense.

Please state clearly that the ToF method uses the SPB approach.

Please clarify the range of angles of the P2T 10MV beams for experiments in Fig 2 and 3, similarly to how you describe it for Fig.4.

You describe the complex dependency of CT contrast formation (Z^3 , shell energy edges..) as a problem, but in my opinion it is rather a good feature, as it greatly enhances image contrast. Please tame down the presentation of the complex CT interactions being a problem.

Reviewer #2 (Report for the authors (Required)):

See **Appendix 2**.

Reviewer #3 (Report for the authors (Required)):

Overall it seems that the authors have addressed the comments of the reviewers quite thoroughly with many modifications to the paper that make it more clear, precise, and readable. I only have a few minor comments:

* Both reviewers 2 and 3 asked about the quality of the FBP result in Fig. 2. I think perhaps this should be addressed in the paper. The TV-min image could be included in Fig 2, or the fruitlessness of regularisation techniques to improve the result should be mentioned.

* I think I now understand that you are saying that P2T is similar to CT in terms of WHAT it is imaging. I accept that position. I guess it is most similar to X-ray fluorescence CT (XRF CT). I think of P2T as similar to PET in terms of HOW it is imaging, i.e., they are both emission tomography, requiring correction for reabsorption of the emitted radiation by the sample itself (CT does not require that), they are both measuring the X-rays emitted from positron annihilation (CT is measuring incident X-rays not attenuated by Photoelectric absorption and Compton Scattering), the only difference is the source of positrons; one is from radioactive tracers, one is stimulated by an X-ray beam with energy $> 1.022\text{MeV}$. Perhaps this WHAT/HOW distinction could be made clear in the introduction?

* Regarding the phase-contrast techniques for enhanced soft tissue differentiation, it is my understanding that pre-clinical systems have been developed that use talbot-lau grating interferometry to remove the coherence restrictions. I think it is worth mentioning even with just a sentence to show that you are aware of this alternative.

* Should your response/graphs to my question about lines 84-85 on positron travel distance be included in the paper? This seems important...

* Should your response to my question about lines 159-162 about ρ/Z be included in the paper? This seems to be an important assumption that ρ is known...

* My question about line 234-235, was an indirect way to say that you should rephrase/expand the sentence to explain that you are assuming a certain measured signal and, if the detector is 100% efficient, that corresponds to a certain dose... can you please do so?

Wed 13 Oct 2021

Decision on Article nBME-21-0210B

Dear Dr Sheng,

Thank you for your revised manuscript, "Pair production tomography". Having consulted with the original reviewers (whose comments you will find at the end of this message), I am pleased to write that we shall be happy to publish the manuscript in *Nature Biomedical Engineering*.

We are now performing detailed checks on your paper and will send you a checklist detailing our editorial and formatting requirements in due course. Please do not upload the final files until you receive this additional information from us.

Best wishes,

Rosy

Dr Rosy Favicchio
Senior Editor, Nature Biomedical Engineering

Reviewer #1 (Report for the authors (Required)):

The authors did a great job addressing all questions and concerns, and all my follow-up questions (regarding confusion of the type of excitation in different imaging methods) were in fact answered elsewhere in the latest authors' response letter. I'm happy to recommend the manuscript for publication, and congratulate the authors for this innovative work.

Reviewer #2 (Report for the authors (Required)):

In this 2nd revision, the authors have greatly improved the clarity and focus of the paper. I have no further concerns.

Just as a note to the authors, the reason that your mean energy for the CT spectrum is around 50 keV is that you are assuming a very soft spectrum (4.3 mm Al filtration), almost radiographic-like. Modern CT systems have a typical HVL of around 10 mm Al.

Thank you for this interesting paper. I look forward to following future work in P2T.

Reviewer #3 (Report for the authors (Required)):

The authors have made very thorough and well considered responses to all of the reviewers comments. Through substantial modifications and additions made to the manuscript, most of these responses have ended up in the paper. I think the review process has achieved its purpose and has resulted in

(i) a clearer presentation of the various excitation and imaging schemes,
(ii) the advantages / disadvantages and opportunities of P2T are now more clear, and
(iii) the simulations are now precisely presented enabling readers to repeat these simulations and confirm the results.

I hope the authors agree that the manuscript is now much stronger than the original submission.

I recommend this paper for publication.

Appendix 1

Key results

This paper shows, through Monte Carlo simulations, the potential for performing CT imaging during high-MeV photon radiation therapy. The concept is very exciting and to my knowledge, novel. The authors describe the background of conventional X-ray CT and the photon interactions responsible for generating the measured signal (attenuation). To facilitate *in vivo* dose measurements during the radiation treatment process, they propose to use the treatment beam as the radiation producing device for image acquisition. They explain how treatment photons with energies > 1.02 MeV undergo pair production (production of an electron and positron). This process naturally occurs as part of the patient irradiation. Their novel concept is to exploit the pair of 511 keV photons that result from a pair production event using a ring of detectors – just like those used in PET imaging – and create a tomographic image from collected data. They detail the advantages of their technique and show images created with 3 different acquisition/reconstruction approaches. The images directly correlate with the dose distribution, which is arguably its greatest strength. They provide an example based on a patient data set. The concept is essentially PET imaging without introduction into the patient of a positron-emitting radioisotope. Instead, the high energy treatment beam creates the electron/positron pair inside the patient. General tomography is then performed, without localization of the signal to any specific tracer uptake. It was a pleasure to read this very approachable and exciting article.

Validity

I find the physical principles related to their approach to be correctly stated and a valid premise for their results and conclusions. They offer to provide all the needed data and computer code for others to replicate their work, which is a testament to their confidence in their results. Unfortunately, insufficient detail is given in a number of cases for a skilled reader to reproduce the exact data shown in this work (e.g., SPB dimensions, Gaussian kernel parameters, ramp filter details). Discussion of the calibration used to determine absolute dose is needed in lines 457 and following.

The Monte Carlo and image reconstruction approaches are straightforward, and I see no obvious concerns in their methodology. My impression is that they have supported their premise that pair production events can be used to produce tomographic images with voxel values directly proportional to density, atomic number, and absorbed dose.

Significance

The significance of gathering CT data during the treatment process is extremely high as this will facilitate image guided radiation therapy and positioning verification. That the images can be calibrated in terms of deposited dose is a game changer in terms of treatment monitoring and adaptation. With this information, the treatment can be updated daily as the tumor shrinks or the patient position changes. Further, it provides data in terms of actually delivered dose to the target and surrounding tissues for use in patient outcome studies of treatment efficacy and side effects. If these data are used to their fullest potential, optimized delivery methods can be confirmed on every patient to avoid treatment failure or damage to adjacent tissues due to inaccurate dose delivery. The 6-day computation time for the dose image, however, and other implementation issues will of course need to be addressed for experimental studies and impose some near-term practical limitations. However, this *in silico* study demonstrated nicely the potential benefits of their approach.

I did a brief literature search to see if anything like this has been reported and found a few references that may be related. It seems that the use of pair production and annihilation photons may have been suggested in the field of proton and ion therapy, but no concrete demonstration performed. I also found references to stimulated emission tomography, which would describe the idea. Those papers were not applied to radiation therapy. My search, however, was of limited scope and I could access the full text, only the abstracts, of the following.

- Tavora, L.M.N., Morgado, R.E., Estep, R.J., Rawool-Sullivan, M., Gilboy, W.B. and Morton, E.J., 1998. One-sided imaging of large, dense objects using 511-keV photons from induced-pair production. *IEEE Transactions on Nuclear Science*, 45(3), pp.970-975.
- Jakeman, E. and Rarity, J.G., 1986. The use of pair production processes to reduce quantum noise in transmission measurements. *Optics communications*, 59(3), pp.219-223.
- Rohling, H., Golnik, C., Enghardt, W., Hueso-González, F., Kormoll, T., Pausch, G., Schumann, A. and Fiedler, F., 2015. Simulation Study of a Combined Pair Production–Compton Camera for In-Vivo Dosimetry During Therapeutic Proton Irradiation. *IEEE Transactions on Nuclear Science*, 62(5), pp.2023-2030.

Data and methodology

The results with the fast detectors (time of flight, TOF) and volume acquisition are very nice, but the practicality of 1440 of these 20 ps detectors is difficult to envision at the current time. What technical advances in detector technology are needed to achieve commercial viability? The authors note that 20 ps is consistent with PET detector technology development roadmaps, but state that considerable engineering will be required to achieve this; what timeframe do they think realistic?

The scanning beam approach to data acquisition is suggested as an alternative to requiring extremely high detector temporal response times (e.g., 20 ps). The practicality of scanning photon beam (SPB) treatment protocols should be discussed. What would that look like in terms of extending treatment times? The results are nice, but is this likely to be clinically viable? They note that current linac multi-leaf collimators could be used to perform the pencil beam scanning, but to what extent will that lengthen the treatment time relative to volume imaging. In the simulation, what was the dimension of the SPB? That is an important parameter as it is a key determinant of spatial resolution and treatment time.

The authors mention on line 121 that they consider both 300 ps and 20 ps, which led me to anticipate seeing an evaluation of TOF at 300 ps; it is not clear in the TOF results which detector was used; Table 1 gives the needed information but in an indirect fashion. Line 121 should be reworded to be explicit regarding which method was evaluated at which detector speed. If the 20 ps detectors are not likely to be achievable in the short term, what does TOF look like for 300 ps? Is that even possible? Line 131 implies that it is not. Please help the reader less familiar with TOF to be aware of this, and to give them a sense of immediate practicality. That is, if a manufacturer would build and put this detector ring in a linac treatment room next year, would this be implementable with either SPC or TOF (w/ 20 ps)?

The filtered backprojection (FBP) reconstruction from a volume acquisition is of quite poor image quality. Does the reconstruction tool kit that they used from Univ of Michigan offer

iterative reconstruction, which would look much better? Newer CNN-based image denoising, deconvolution, and reconstruction methods are available and more being developed, so if SPB and TOF are not practical, volume acquisition (best for treatment time) combined with advanced reconstruction and denoising is likely to yield images much closer in quality to SPB and TOF. That should be pointed out. Also, the use of a plain ramp filter, without benefit of apodization, is a weakness of the FBP data set. A number of refinements are needed to optimize FBP reconstructions that were not taken here. I do not think that the results shown adequately demonstrate how good traditional CT reconstructions can be and certainly aren't state of the art. Collaboration with CT image reconstruction experts is needed, in my opinion, to improve the results with the most straight-forward data acquisition approach.

I do not have expertise in Monte Carlo methods, just a general understanding. The methods seem generally appropriate, however they made a number of simplifying assumptions. A discussion of the magnitude of the expected impact of those real-world effects on the results should be included (non-ideal beam geometry, non-ideal detector absorption and localization, parallax, etc.). The suggestion to adopt a whole-body PET approach is impractical given the need for the treatment beam to have broad access to the patient and the cost, which would likely be prohibitive for the foreseeable future.

Analytical approach

With the exception of the FBP image reconstruction, the analytical approaches used are appropriate. These are primarily described in the methods section.

Suggested improvements

Additional work to improve the FBP images is strongly recommended, otherwise the impression is that only SPB and TOF options are viable, even though each face practical limitations. The benefits of the straightforward data acquisition approach justify greater effort on the FBP images.

Clarity and context

I find the manuscript to be quite readable and to convey the work clearly. I found some of the early content a bit basic (e.g., the mechanisms for non-pair-production photon interactions), but they will likely be of value for scientists not versed in radiological physics.

References

A number of references are given to textbooks, which is not particularly useful to a reader without specific page numbers. Readers are unlikely to read an entire textbook to find support for a given statement in the paper. I don't think that references are needed for fundamental facts such as the mechanism for pair production, etc.

Major comments

1. My major concern is the simplistic FBP reconstruction and the poor quality that results.
2. Line 19: A claim is made regarding "typical imaging dose", but no dose values or criteria were discussed. How is that relevant for the therapy application anyway, where the treatment dose determines the beam fluence? What dose level do the simulations correspond to in patients? Do the authors foresee an application where the MeV photons are used only for imaging and not for simultaneous treatment?

3. Please correct the assertion that the PP cross section is proportional to Z (line 87). As the authors state on line 418, the PP cross-section is proportional to Z^2 . As shown on line 422, it is the attenuation coefficient that is proportional to Z.

Minor comments

1. Line 11: FBP is not an acquisition method. It is a reconstruction method. SPB describes both acquisition and reconstruction approaches, as does TOF. Please reword.
2. Line 15: "as few as a single" – suggest rewording for clarity. Something like "The ability to perform tomography with just one highly-truncated x-ray beam." Also, this assertion is not supported by the study. The minimum beam approach shown is 2 view.
3. Line 31: lung cancer screening is a poor example. It is just a subset of routine imaging of the chest. CT revolutionized medicine from the very start simply by providing cross-sectional images of patient anatomy in the head. Stay general and don't try to give an examples, as there are 100s of previously impossible things that CT allows.
4. P^2T is awkward to speak "P squared T" and doesn't flow easily like CT, PET, MRI. I suggest simply PPT. Easier to type also (not needing a superscript every time that one writes it).
5. The background of the Figures in 1a should be white. This will increase clarity and match the other figures more nicely.
6. Line 114: Why is such a large diameter detector ring assumed. This seems to be a detriment for image quality due to the r^2 loss of fluence.
7. Table 1 should be explained in greater detail and all abbreviations spelled out in the legend.
8. Line 200: ρ is defined above as density. Please clarify whether mass density, as is used in CT, or electron density, as used in radiation oncology.
9. The results of Figure 3b deserve discussion. For bone, traditional CT is best. How will that impact dose calculations? FBP is frequently the best (in terms of contrast), making optimization of the FBP (or other) reconstruction of the volume acquisition data even more important.
10. Figure 4a: Which P^2T image was used to calculate the dose image?
11. Figure 4a: What are the units on the intensity grey scale? Why is that even necessary for the grey scale images?
12. The question regarding funding source on the manuscript processing website says that there are no funding sources, yet the paper lists 2. Please reconcile.

Appendix 2

Reading the responses to all of the referees, my overall enthusiasm is considerably decreased as it is now apparent to me that the authors are suggesting a proposed utility as a primary imaging modality (what they refer to as the low dose imaging scenarios). I find that to be ill-informed of clinical realities and to not fully appreciate the value of the “high dose” radiation therapy dose monitoring application, which I found to be extremely exciting.

I lack enthusiasm for any purely imaging application due to the following:

1. The imaging method depends on a source of photons with energies > 1.022 MeV. The broad use of CT in all medical setting requires efficiency in terms of cost of equipment, cost of facility requirements such as shielding and power, cost of space requirements, room throughput, service expenses, and more. Linacs are expensive, require expensive shielding and building footing, take up lots of space, and require service and calibrations not required of x-ray tube based imaging systems. The cost for any linac-based “imaging system” is a non-starter. This technique only makes sense in the presence of an existing MeV x-ray source.
2. Outside of the radiation therapy environment, the ability to resolve material atomic number adds value to diagnostic utility of CT scanning for only some specific situations. For those situations (e.g., separating iodine and bone), dual-energy CT and photon-counting-detector CT are capable of addressing most of these needs. Where they are limited, such as in evaluation of tendons and ligaments, MRI provides the needed capabilities. The primary value of CT is in fast, robust, high-spatial resolution imaging of anatomic detail. The inherent spatial resolution limits noted in the responses (e.g., 3 mm in the best case of 20 ps resolution detectors for TOF image reconstruction, 45 mm in the case of SPB, which is more practical with modern technology) is already far inferior to CT, which is below 150 micron on the newest high-end CT systems and at or below 1 mm in even low-end systems, which can cost under \$200,000. The utility of P2T for imaging effective atomic number outside of radiation therapy treatment planning for proton therapy or radiation therapy dose monitoring pales in comparison to the overwhelming value of the sharp definition of anatomic detail. The blurriness of the P2T images in Figure 3 compared to the sharpness of the CT image already points to this weakness of P2T – and this was in an ideal environment with a point source and ignoring non-ideal detector properties. Also very concerning is the information provided in the response to reviewer #3 regarding the range of uncertainty due to positron travel: “The median of positron travelling distances before annihilation is 4.6 mm.” The tail of the distribution given extends out to 25 mm. While the authors state that deconvolution with a simulated distribution of positron travel distances can improve resolution, it is not clear by how much and at what expense in noise (deconvolution increases image noise). With such fundamental limitations on spatial resolution, the idea of P2T serving as an imaging modality seems to me a solution in search of a problem. The clear problem that it can solve is in the radiation therapy context and this needs to be recognized and made clear in the paper.
3. The simulations – again using ideal source geometries and detectors – used a dose of 36 mGy in a 20x24 cm phantom - assuming a 100% efficient detector. That dose is already a factor of more than 3 higher than typical doses using low end scanners. In such a small patient, 12 mGy would be considered at the upper end of clinical doses. With modern technologies, doses can be in the range of 4-6 mGy in the abdomen and even lower in the thorax. Thus, P2T technology, at some future time when 20 ps temporal resolution detectors are possible, would be at a considerable dose disadvantage to x-ray tube based CT, which will continue to evolve as more dose reduction methods are adopted commercially.

4. The elements used in Figure 2 to show linearity with atomic number have no real significance in medical imaging, with the exception of Iodine and Barium, which are easily distinguishable in medical imaging due to their method of injection or ingestion. The approval of new contrast agents is a billion dollar prospect and thus it is unlikely that these rare earth materials would be added to the very limited menu of contrast agents. Even though approved for use in humans, and extensively studied and improved when safety issues did arise, there is almost no interest in adopting Gadolinium in CT, even for spectral applications. Thus the linearity of signal to Z offers no compelling benefit for purely imaging applications (unrelated to radiation therapy).
5. In figure 3, none of the P2T techniques differentiate liver from muscle from water – the effective atomic numbers are too similar. Even bone has lower contrast from water in P2T than in CT. The only materials that show improved contrast relative to water are breast tissue and adipose tissue (where there is a small difference in atomic number). The CT simulation is for some reason not a good indication of CT's ability to image fat. Clinically, fat is extremely easy to differentiate from other forms of soft tissue. I see no benefit in P2T if it can give more contrast to fat, which is already well seen, but has no contrast from muscle or liver, which are also already well seen and are essential to be seen.
6. The answer to referee #1's question about scan time is 28 s, although they indicate this is only an "order of magnitude" estimate. It is not clear how much imaging volume this would include, which presumably would depend on the number of detector rows (and their resolution) longitudinally. Current systems can scan an adult from head to toe in around 20 s. Some systems can do this in under 5 s. Short of whole-body PET systems, which are out of the question from a cost point of view, scan time would be a major issue for P2T.

Additional comments:

7. The limits to resolution for SPB (45 mm) should be noted, as well as the limits to resolution due to travel of the positron from its point of creation before its annihilation.
8. Better differentiation between "approaches," "acquisition methods," and "acquisition and reconstruction methods" are needed. Even after multiple reads, it still gets confusing. An examples is in the legend for Table 1. What is the difference in imaging methods and imaging approaches? I found the phrase "volume imaging" or "full FOV" to be clear (used in Figure 4). Table 1 needs a matrix to clearly show which combinations of irradiation and which reconstruction approaches are investigated. I also don't understand why attenuation correction is not applicable to SPB.
9. Lines 11-14 state: We studied three P2T acquisition methods: a scanning pencil beam (SPB) and two full-field illumination methods, one with temporal resolution sufficient to identify coincident photon-pairs and one with higher temporal resolution sufficient for photon time-of flight (TOF) estimation. However, Figure 4 shows results for full and partial FOV, implying that SPB was also studied for full-field illuminations and FBP and TOF also studied with partial view illumination. Please increase the clarity as to which field illumination technique and reconstruction approach were studied, since Figure 4 implies all combinations were but the text gives only specific combinations.
10. Lines 76-77: Different from a previous work using pair production for one-sided point material detection, ... please clarify the meaning of one-sided point detection.
11. Lines 94-96: An X-ray beam typically used for radiotherapy introduces the pair production photons This is a confusing phrase. Pair production is an interaction, not photons. There are "pair production" electrons and positrons as they result from the interaction. Please differentiate that the photons interact via pair production to create the

electron-positron pair. This would be a good place to discuss the issue of positron travel prior to annihilation. The coincident photons are annihilation photons. Please help the reader to keep these straight.

12. Line 105: Insert the word “imaging” in front of “approaches.” This is an example of where it seems 2 different imaging “approaches” are studied but the abstract talks about three acquisition methods. Confusing. Lines 132-133 talk about three imaging acquisition and reconstruction methods but you are really laying out the 3 reconstruction methods (FBP, SBP, and TOF). Some of the confusion is that SBP is a unique acquisition method that enables a unique reconstruction method. It seems to me that volume or full-field irradiation/illumination (acquisition) is used with FBP and TOF reconstruction, where TOF is only done assuming 20 ps detector but FBP assumes 300 ps detectors (as does SPB). Table 1 should lay out acquisition and reconstruction and detector assumptions separately and you need to stay with the same terminology throughout that clearly defines the combination you are referring to. The terms approaches and methods are not specific, but the terms acquisition and reconstruction are.
13. Line 191: Gadolinium ... its K-edge energy at 50 keV is closer to the peak of the 120 kVp CT spectrum. Consequently, the CT contrast of Gadolinium is substantially higher than other materials. The k-edge at 50 keV is closer to the **effective energy** of a 120 kVp CT beam (i.e., 70 keV), not the peak energy. I’m not sure this reasoning is correct, however, seeing as the k-edge of Lu is 63 keV and Ta is 67 keV, both are which are closer to 70 keV than 50 keV is. The signal amplitude in CT is not just about attenuation coefficients. The energy response of the detector comes into play. Around Line 180, where the CT simulation information is given, the detector material, density and thickness are not detailed. The use of any collimator grids is not noted (which are essential in CT imaging to decrease scatter and improve contrast resolution). The beam filtration is not given. CT beam filtration is very different than what is used for 120 kVp in radiographic and fluoroscopic applications. Without these key details, the work presented cannot be replicated and the realism of the CT simulation cannot be assessed. All of the MC simulation experimental details should be summarized in a table. Include an appendix if necessary.
14. Line 243: The phrase “ultra-low-dose” is hyperbole. Give a numerical estimate based on the literature. Be aware that CT scanners cannot deliver infinitely low doses. Doses are bounded by realistic tube and generator specifications.
15. Line 381: What is “perfect beam shaping?” Beam shaping in what respect? Spectral? Spatial? Point focal spot? Shielding is mentioned. What impact would that have on the results of the simulation? Presumably degradation in spatial resolution due to increased spread of the focal spot?
16. Line 385: Electronic noise and detector cross-talk definitely will (not may) impact SNR.
17. Unfortunately, insufficient detail is still a problem.
18. The results with the fast detectors (time of flight, TOF) and volume acquisition are very nice, but the practicality of 1440 of these 20 ps detectors is difficult to envision at the current time. What technical advances in detector technology are needed to achieve commercial viability? Authors: Cherenkov light is a potential approach to achieve 20 ps detector time resolution. A simulation study reported an ideal photo-detector time response at 22 ps². But it is difficult to achieve both good detection efficiency and time resolution using Cherenkov light. Only 6% single side detection efficiency was reported³. Achieving high time resolution without sacrificing detector efficiency is required for commercial viability. The discussion should more clearly emphasize the long road to technically achieve 20 ps resolution as many readers of this journal will not be familiar with the limitations of detector technologies.

19. The authors note that 20 ps is consistent with PET detector technology development roadmaps, but state that considerable engineering will be required to achieve this; what timeframe do they think realistic? Authors: 20ps PET TOF detector is still under active research development⁴. Its commercialization is pending upon overcoming many technical difficulties. We do not have enough information to estimate a timeline. Same point of above. Enthusiasm about this paper depends in a large extent to practicality, especially since it aims to offer an alternative to CT, which is already well developed and ubiquitously deployed.
20. The scanning beam approach to data acquisition is suggested as an alternative to requiring extremely high detector temporal response times (e.g., 20 ps). The practicality of scanning photon beam (SPB) treatment protocols should be discussed. What would that look like in terms of extending treatment times? The results are nice, but is this likely to be clinically viable? They note that current linac multi-leaf collimators could be used to perform the pencil beam scanning, but to what extent will that lengthen the treatment time relative to volume imaging. Authors: SPB is viable with current technology. A typical multi-leaf collimator (MLC) has 5mm leaf width (pencil-beam width). The beam width can be further reduced to 2.5mm using the dual-layer MLC^{5,6}. Treatment machines with dual-layer MLC are already commercially available.

The total imaging time of SPB using MLC includes beam-on time and leaf-moving time. The SPB beam-on time is comparable to that of volume imaging (20s to 30s), which is limited by the maximum photon flux due to a finite detector dead time. The leaf-moving time is limited by the maximum leaf-moving speed. For example, assuming the maximum leaf-moving speed of 2.5cm/s, it takes 50s leaf-moving time to scan a 5cm-by-5cm imaging field-of-view (2s per row and 25 rows in total). Is the 50 s lead moving on top of the data acquisition/collection time? This journal has broad readership and the authors cannot assume familiarity with MLC technology. The introduction provided very broad, even somewhat basic, information but in other sections specialty expertise is assumed. Even if 50 s is the total time, that is completely untenable for patient imaging, which much be performed within one breathhold (20 s is an upper limit for most patients). Plus, this covers only a small in-plane field of view (5 cm x 5 cm). What is the coverage in the long axis?

21. Please make these points more clear in the manuscript:
- “For a single annihilation event, with imaging dose in the diagnostic range, TOF of 300 ps results in a 45 mm range, which in itself is inadequate resolution. However, when a large number of annihilation events are recorded in radiotherapy with 2 orders of magnitude greater dose, useful images can be reconstructed in a statistical manner using FBP as shown in Figure 5.”
 - SPB is readily available using current technology.
 - In the low dose imaging setting, FBP images with broad beam irradiation are included in the manuscript as a reference. SPB and TOF will likely be the method of choice in the low dose mode. In the high-dose therapeutic mode, however, FBP still results in highly usable images (Figure 5).
 - We presented three imaging applications and one radiotherapy application in this study. The three P2T imaging applications use MeV photons only for imaging and not for simultaneous treatment. --- Please enumerate the clinical applications more clearly. I don't see three of them. Also put the potential clinical value into perspective. Also note that region-of-interest CT (“truncated imaging”) has been demonstrated for well over a decade but there is no clinical interest as the applications would be so limited. Part of the value of CT is that it identifies non-expected findings. By covering a large volume, it quickly gives information about the entire body region, and that is one of its key value

propositions compared to MRI and US, which both require more focused imaging of targeted regions.

- e. The three P2T images have the same imaging dose.
- f. However, since the cost is irrelevant to the current manuscript, we removed the corresponding reference in the revised manuscript. – I don't think cost is irrelevant. Not only are the detectors much more expensive, the need for a megavoltage source is vastly more expensive. This paper demonstrates theoretical potential, and implies to the reader that there may be value in such an imaging paradigm. In the absence of practical pure imaging applications, due to the points discussed above, this is unrealistic. I hope that the authors will come back with a revision that hits a home run with the radiation therapy potential applications and avoids suggesting highly unlikely pure imaging applications.
- g. If the detector efficiency is 10%, only 10% of the coincident photon pairs will be collected as data. The 10% efficiency we assume in the study is conservative and should set a lower performance bound. The dose that you estimate is based on 100% detector efficiency and here you note 10%, implying to me that your dose estimate is not very realistic.

Rebuttal 1

Dear Editor and Reviewers,

We thank you for your positive evaluation and helpful comments, based on which we have revised the manuscript. Your comments are individually responded to as follows with corresponding highlighted changes in the revision.

Sincerely yours

Authors.

Referee #1:

The authors describe a very interesting and novel tomography imaging method that utilizes the density of megavoltage X-ray beam-induced pair production events to generate true atomic density distribution maps. The study is purely theoretical, however it addresses many practical questions by approximating the limitations of real detectors and doses. The authors demonstrate three different reconstruction approaches, which cover different applications and detector types (Time-of-flight, pencil beam scanning, etc.), as well as three different real-world scenarios for high-Z CT, tissue CT, and in vivo beam dosimetry. The manuscript is exceptionally well written, clear, and addresses reader's questions seamlessly. This work has the potential to be very influential, provided that the approximations here are representative. I am not 100% convinced the signal to noise will be satisfactory with current setups and doses usual for standard CT, but it will be exciting to see this application unveil its benefits as the technology advances. The theoretical explanation is simple yet clean and understandable, and I did not find any issue with it. I am enthusiastic about this work and its merit for Nature Biomedical Engineering, and have only a few methodology questions and several minor comments about the figures.

CT requires much simpler, charge-integrating detector in comparison to PET. What is the required dose rate (and therefore real scan time) necessary to produce the demonstrated figures, given a certain non-zero dead time of photon counting PET detectors?

The answer to this question depends on specific detector configuration and setup. Here we provide a general estimate. In order to separate pair production photons from Compton scatter photons, our simulation suggested that the dose rate should be no more than 100Gy/min. The results presented in this study assumed 0.1Gy/min dose rate, which corresponds to 28s scan time. The photon rate at each detector is 4000 photons/s, which is substantially lower than the published photon count rate limit at 200 000 events/s¹ due to detector dead time. The order of magnitude estimates indicate flexibility to acquire images at a range of speeds and timeframes.

In order for energy discriminator to work efficiently, the photon counting rate shall not exceed say 1-10% of theoretical counting rate to prevent event pile-up. Considering the strong, omnidirectional scatter present due to Compton and other mechanisms, I can imagine the detectors may be flooded

with this background signal, preventing the energy filtering and coincidence detection to work. Can the authors shed light on the discrimination of pair production photons and all the background? This could be included as a figure in a form of time-resolved histogram, similar to that in Fig 1, but with a representative PET detector and the selected object under scan (for example, the one used later in manuscript). This can definitely grow out of the scope of this manuscript very fast, but I believe the authors should take this into consideration and demonstrate it, as it seems to be pivotal for the success of this method.

Assuming the theoretical photon count rate is 200 000 events/s, then the photon-counting rate cannot exceed 20 000 events/s (50 ms per event) to prevent event pile up. The following figure is a time-resolved histogram showing the time difference between adjacent detected signals for a single detector element. The majority of events (75%) are separated from the last detection by more than 50 ms. The ratio can be further improved if using a lower dose rate or a shorter detector dead time.

Fig. 1 c)-e) is missing axis labels - even though it is explained in the caption, it would add clarity to annotate them in figures.

Done

Likewise, the total dose used to form Figures 2 and 3 could be mentioned in caption, so it is immediately clear.

Done

The color bars are occasionally missing a label, and the text size may be too low in final format.

Done

I will gladly recommend the manuscript for publication once the technical comments and stylistic issues are addressed.

Reviewer #2:

Key results

This paper shows, through Monte Carlo simulations, the potential for performing CT imaging during high-MeV photon radiation therapy. The concept is very exciting and to my knowledge, novel. The authors describe the background of conventional X-ray CT and the photon interactions responsible for generating the measured signal (attenuation). To facilitate in vivo dose measurements during the radiation treatment process, they propose to use the treatment beam as the radiation producing device for image acquisition. They explain how treatment photons with energies > 1.02 MeV undergo pair production (production of an electron and positron). This process naturally occurs as part of the patient irradiation. Their novel concept is to exploit the pair of 511 keV photons that result from a pair production event using a ring of detectors – just like those used in PET imaging – and create a tomographic image from collected data. They detail the advantages of their technique and show images created with 3 different acquisition/reconstruction approaches. The images directly correlate with the dose distribution, which is arguably its greatest strength. They provide an example based on a patient data set. The concept is essentially PET imaging without introduction into the patient of a positron-emitting radioisotope. Instead, the high energy treatment beam creates the electron/positron pair inside the patient. General tomography is then performed, without localization of the signal to any specific tracer uptake. It was a pleasure to read this very approachable and exciting article.

Validity

I find the physical principles related to their approach to be correctly stated and a valid premise for their results and conclusions. They offer to provide all the needed data and computer code for others to replicate their work, which is a testament to their confidence in their results. Unfortunately, insufficient detail is given in a number of cases for a skilled reader to reproduce the exact data shown in this work (e.g., SPB dimensions, Gaussian kernel parameters, ramp filter details). Discussion of the calibration used to determine absolute dose is needed in lines 457 and following.

Details on the parameters used in this study were added to the corresponding texts in the method section. The radiotherapy treatment dose and imaging dose are acquired simultaneously with the PPTI signals using Monte Carlo simulation. Therefore, no calibration was applied.

The Monte Carlo and image reconstruction approaches are straightforward, and I see no obvious concerns in their methodology. My impression is that they have supported their premise that pair production events can be used to produce tomographic images with voxel values directly proportional to density, atomic number, and absorbed dose.

Significance

The significance of gathering CT data during the treatment process is extremely high as this will facilitate image guided radiation therapy and positioning verification. That the images can be calibrated in terms of deposited dose is a game changer in terms of treatment monitoring and adaptation. With this information, the treatment can be updated daily as the tumor shrinks or the patient position changes. Further, it provides data in terms of actually delivered dose to the target and surrounding tissues for use in patient outcome studies of treatment efficacy and side effects. If these data are used to their fullest potential, optimized delivery methods can be confirmed on every patient to avoid treatment failure or damage to adjacent tissues due to inaccurate dose delivery. The 6-day computation time for the dose

image, however, and other implementation issues will of course need to be addressed for experimental studies and impose some near-term practical limitations. However, this *in silico* study demonstrated nicely the potential benefits of their approach.

I did a brief literature search to see if anything like this has been reported and found a few references that may be related. It seems that the use of pair production and annihilation photons may have been suggested in the field of proton and ion therapy, but no concrete demonstration performed. I also found references to stimulated emission tomography, which would describe the idea. Those papers were not applied to radiation therapy. My search, however, was of limited scope and I could access the full text, only the abstracts, of the following.

Tavora, L.M.N., Morgado, R.E., Estep, R.J., Rawool-Sullivan, M., Gilboy, W.B. and Morton, E.J., 1998. One-sided imaging of large, dense objects using 511-keV photons from induced-pair production. *IEEE Transactions on Nuclear Science*, 45(3), pp.970- 975.

Jakeman, E. and Rarity, J.G., 1986. The use of pair production processes to reduce quantum noise in transmission measurements. *Optics communications*, 59(3), pp.219- 223.

Rohling, H., Golnik, C., Enhardt, W., Hueso-González, F., Kormoll, T., Pausch, G., Schumann, A. and Fiedler, F., 2015. Simulation Study of a Combined Pair Production– Compton Camera for In-Vivo Dosimetry During Therapeutic Proton Irradiation. *IEEE Transactions on Nuclear Science*, 62(5), pp.2023-2030.

We read the papers found by the reviewer with great interest. The paper by Tavora et al. describes the detection of high Z materials with MV gamma-ray-produced pairs in the sample. The detection was based on the same physical process of pair production for a very different application, where one only had access to one side of the sample. In this application, only point measurements were obtained, in contrast to our 3D tomographic image formation. On the other hand, the paper provides evidence that from a physics perspective, our idea is feasible! We cited the paper in the introduction as follows:

“Different from a previous work using pair production for one-sided point material detection (Tavora et al.), 3D P2T imaging significantly expands the capacity for medical applications.”

The second paper by Jakeman et al., however, describes a very different physical process. In this case, the source itself was a positron emitter. One of the coincidence photons was used to expose the sample before collecting by the detector, while the other was collected by the detector directly. By doing so, for the same imaging dose, the signal-to-noise ratio was significantly improved, compared with single-photon attenuation X-ray imaging. The idea is ingenious but evidently different from our study. The subject of the third paper by Rohling et al. was proton therapy, which generates a broad spectrum of prompt gamma-rays with a substantial high-energy component for pair production. However, different from our study, for *in vivo* prompt gamma-rays imaging, the pairs produced in the patients were *not* measured. Instead, the pairs were produced in the imaging device called pair production camera. Therefore, this is also different from our study.

Data and methodology

The results with the fast detectors (time of flight, TOF) and volume acquisition are very nice, but the practicality of 1440 of these 20 ps detectors is difficult to envision at the current time. What technical advances in detector technology are needed to achieve commercial viability?

Cherenkov light is a potential approach to achieve 20 ps detector time resolution. A simulation study reported an ideal photo-detector time response at 22 ps². But it is difficult to achieve both good detection efficiency and time resolution using Cherenkov light. Only 6% single side detection efficiency was reported³. Achieving high time resolution without sacrificing detector efficiency is required for commercial viability.

The authors note that 20 ps is consistent with PET detector technology development roadmaps, but state that considerable engineering will be required to achieve this; what timeframe do they think realistic?

20ps PET TOF detector is still under active research development⁴. Its commercialization is pending upon overcoming many technical difficulties. We do not have enough information to estimate a timeline.

The scanning beam approach to data acquisition is suggested as an alternative to requiring extremely high detector temporal response times (e.g., 20 ps). The practicality of scanning photon beam (SPB) treatment protocols should be discussed. What would that look like in terms of extending treatment times? The results are nice, but is this likely to be clinically viable? They note that current linac multi-leaf collimators could be used to perform the pencil beam scanning, but to what extent will that lengthen the treatment time relative to volume imaging. In the simulation, what was the dimension of the SPB? That is an important parameter as it is a key determinant of spatial resolution and treatment time.

This study assumed 2mm beamlet width for SPB.

SPB is viable with current technology. A typical multi-leaf collimator (MLC) has 5mm leaf width (pencil-beam width). The beam width can be further reduced to 2.5mm using the dual-layer MLC^{5,6}. Treatment machines with dual-layer MLC are already commercially available.

The total imaging time of SPB using MLC includes beam-on time and leaf-moving time. The SPB beam-on time is comparable to that of volume imaging (20s to 30s), which is limited by the maximum photon flux due to a finite detector dead time. The leaf-moving time is limited by the maximum leaf-moving speed. For example, assuming the maximum leaf-moving speed of 2.5cm/s, it takes 50s leaf-moving time to scan a 5cm-by-5cm imaging field-of-view (2s per row and 25 rows in total).

The authors mention on line 121 that they consider both 300 ps and 20 ps, which led me to anticipate seeing an evaluation of TOF at 300 ps; it is not clear in the TOF results which detector was used; Table 1 gives the needed information but in an indirect fashion. Line 121 should be reworded to be explicit regarding which method was evaluated at which detector speed.

We added the following text in the revision:

“Among them, the time-of-flight (TOF) information of the 20 ps detector allows directly locating an annihilation event with 3mm resolution, but the 300 ps detector only has 45mm resolution, which is inadequate for practical applications. Therefore, we did not consider TOF reconstruction for 300 ps detector in this study.”

If the 20 ps detectors are not likely to be achievable in the short term, what does TOF look like for 300 ps? Is that even possible? Line 131 implies that it is not. Please help the reader less familiar with TOF to be aware of this, and to give them a sense of immediate practicality.

For a single annihilation event, with imaging dose in the diagnostic range, TOF of 300 ps results in a 45 mm range, which in itself is inadequate resolution. However, when a large number of annihilation events are recorded in radiotherapy with 2 orders of magnitude greater dose, useful images can be reconstructed in a statistical manner using FBP as shown in Figure 5.

That is, if a manufacturer would build and put this detector ring in a linac treatment room next year, would this be implementable with either SPC or TOF (w/ 20 ps)?

SPB is readily available using current technology. P2T acquired and reconstructed using SPB with 300ps detector should be available earlier than TOF images that require 20 ps detectors.

The filtered backprojection (FBP) reconstruction from a volume acquisition is of quite poor image quality. Does the reconstruction tool kit that they used from Univ of Michigan offer iterative reconstruction, which would look much better? Newer CNN-based image denoising, deconvolution, and reconstruction methods are available and more being developed, so if SPB and TOF are not practical, volume acquisition (best for treatment time) combined with advanced reconstruction and denoising is likely to yield images much closer in quality to SPB and TOF. That should be pointed out. Also, the use of a plain ramp filter, without benefit of apodization, is a weakness of the FBP data set. A number of refinements are needed to optimize FBP reconstructions that were not taken here. I do not think that the results shown adequately demonstrate how good traditional CT reconstructions can be and certainly aren't state of the art. Collaboration with CT image reconstruction experts is needed, in my opinion, to improve the results with the most straight-forward data acquisition approach.

A comparison of FBP and TV reconstruction of P2T images are shown in the figure below. Iterative reconstruction using TV regularization suppresses the image noise, but it does not show additional information that is invisible in the FBP image. In other words, the reconstruction problem is too ill-conditioned to be saved by reconstruction algorithms alone.

Similarly, CNN-based denoising, deconvolution, and reconstruction methods, or FBP with other filters such as hanning filter, are unlikely to produce image quality matching that of SPB and TOF. Improving the image quality will require collecting more photons, either by increasing the fluences (and subsequently imaging dose), or by increasing detector geometrical efficiency. In the low dose imaging setting, FBP images with broad beam irradiation are included in the manuscript as a reference. SPB and TOF will likely be the method of choice in the low dose mode. In the high-dose therapeutic mode, however, FBP still results in highly usable images (Figure 5).

I do not have expertise in Monte Carlo methods, just a general understanding. The methods seem generally appropriate, however they made a number of simplifying assumptions. A discussion of the magnitude of the expected impact of those real-world effects on the results should be included (non-ideal beam geometry, non-ideal detector absorption and localization, parallax, etc.). The suggestion to adopt a whole-body PET approach is impractical given the need for the treatment beam to have broad access to the patient and the cost, which would likely be prohibitive for the foreseeable future.

We added the following text:

“For simplicity, we assumed an ideal point source with perfect beam-shaping. In reality, a shielding structure is required to remove X-ray photons due to source leakage (~1% of primary X-ray photons), so that they wouldn’t interfere with the P2T signals. We also assumed an ideal ring-detector without electronic noise and cross-talks in adjacent detector elements. In reality, the detector response may decrease the image SNR and resolution. In addition, the photon may travel through a few detector modules before generating a signal, causing parallax error.”

Analytical approach

With the exception of the FBP image reconstruction, the analytical approaches used are appropriate. These are primarily described in the methods section.

Suggested improvements

Additional work to improve the FBP images is strongly recommended, otherwise the impression is that only SPB and TOF options are viable, even though each face practical limitations. The benefits of the straightforward data acquisition approach justify greater effort on the FBP images.

As explained, in the low dose imaging mode, without TOF, there is insufficient information for reconstruction of useable images with FBP or other reconstruction methods. In such case, either SPB or TOF are the methods of choice. In the high dose mode, however, usable images can be obtained using FBP with readily available 300 ps detectors.

Clarity and context

I find the manuscript to be quite readable and to convey the work clearly. I found some of the early content a bit basic (e.g., the mechanisms for non-pair-production photon interactions), but they will likely be of value for scientists not versed in radiological physics.

References

A number of references are given to textbooks, which is not particularly useful to a reader without specific page numbers. Readers are unlikely to read an entire textbook to find support for a given statement in the paper. I don’t think that references are needed for fundamental facts such as the mechanism for pair production, etc.

Done

Major comments

1. My major concern is the simplistic FBP reconstruction and the poor quality that results.

We responded to an earlier comment. The quality of FBP in the low dose imaging mode is physically limited. Potential for further image quality improvement using iterative construction is limited. Although machine learning or deep learning methods can potentially improve the reconstructed images, they deviate from the original intent of the manuscript to introduce a novel image formation mechanism. Furthermore, we include the FBP images in the low dose as a reference. SPB and TOF resulting in usable images should be the method of choice here. In the high-dose therapeutic mode, however, FBP still results in highly usable images.

2. Line 19: A claim is made regarding “typical imaging dose”, but no dose values or criteria were discussed. How is that relevant for the therapy application anyway, where the treatment dose determines the beam fluence? What dose level do the simulations correspond to in patients? Do the authors foresee an application where the MeV photons are used only for imaging and not for simultaneous treatment?

We presented three imaging applications and one radiotherapy application in this study. The three P2T imaging applications use MeV photons only for imaging and not for simultaneous treatment. The imaging dose is similar to a typical CT scan, with a maximum point dose at around 3.5cGy. In the radiotherapy application, 2Gy dose was delivered to the patient PTV. We added the dose information to the manuscript for each specific case.

3. Please correct the assertion that the PP cross section is proportional to Z (line 87). As the authors state on line 418, the PP cross-section is proportional to Z^2 . As shown on line 422, it is the attenuation coefficient that is proportional to Z.

Done

Minor comments

1. Line 11: FBP is not an acquisition method. It is a reconstruction method. SPB describes both acquisition and reconstruction approaches, as does TOF. Please reword.

We changed it to:

“We studied three P2T acquisition methods: a scanning pencil beam (SPB) and two full-field illumination methods, one with temporal resolution sufficient to identify coincident photon-pairs and one with higher temporal resolution sufficient for photon time-of-flight (TOF) estimation.”

2. Line 15: “as few as a single” – suggest rewording for clarity. Something like “The ability to perform tomography with just one highly-truncated x-ray beam.” Also, this assertion is not supported by the study. The minimum beam approach shown is 2 view.

We changed it to: “the ability to perform tomography with just one highly-truncated x-ray beam through the region-of-interest (ROI),”

3. Line 31: lung cancer screening is a poor example. It is just a subset of routine imaging of the chest. CT revolutionized medicine from the very start simply by providing crosssectional images of patient

anatomy in the head. Stay general and don't try to give an examples, as there are 100s of previously impossible things that CT allows.

We deleted the example.

4. P2 T is awkward to speak "P squared T" and doesn't flow easily like CT, PET, MRI. I suggest simply PPT. Easier to type also (not needing a superscript every time that one writes it).

We changed the term to P2T, which is easy to type and pronounce yet differentiable from PowerPoint. One could also interpret it as Pair-Production To Tomography.

5. The background of the Figures in 1a should be white. This will increase clarity and match the other figures more nicely.

Done

6. Line 114: Why is such a large diameter detector ring assumed. This seems to be a detriment for image quality due to the r^2 loss of fluence.

We assumed a large diameter detector ring such that it is compatible with radiotherapy machines. For a pure imaging system, we agree that the diameter can be substantially reduced for better signal collection efficiency. In such a case, our current simulation serves as a lower performance bound.

7. Table 1 should be explained in greater detail and all abbreviations spelled out in the legend.

Done

8. Line 200: rho is defined above as density. Please clarify whether mass density, as is used in CT, or electron density, as used in radiation oncology.

We clarified that it refers to the mass density.

9. The results of Figure 3b deserve discussion. For bone, traditional CT is best. How will that impact dose calculations? FBP is frequently the best (in terms of contrast), making optimization of the FBP (or other) reconstruction of the volume acquisition data even more important.

Both CT and P2T have high image contrast on bone structures. The image contrast should be adequate for dose calculation.

We agree in principle for CT reconstruction, FBP is indispensable. However, due to the different image formation mechanisms, the performance of FBP for P2T is limited by physics. It has a larger deviation from ground truth than SPB and TOF. The higher FBP contrast shown in Figure 3 for several structures, including the muscle, liver, and bone, is due to bias and distortion without sufficient photon statistics.

On the other hand, we do not preclude future improvement of FBP using methods such as deep learning but believe that we should save those studies for follow-up research without distracting the readers from the first report of P2T.

10. Figure 4a: Which P2 T image was used to calculate the dose image?

The three P2T images have the same imaging dose. The FBP and TOF assumed different detector time resolution, which do not affect imaging dose. The SPB assumed scanning pencil beam, but the combined x-ray fluence from all pencil beams is the same as volume acquisition.

11. Figure 4a: What are the units on the intensity grey scale? Why is that even necessary for the grey scale images?

The grey scale images are unitless. The intensity values were normalized such that the intensity of water is 1. The grey scale provides a reference for the intensity values of P2T images, which matches with the barplots on Figure 4b.

12. The question regarding funding source on the manuscript processing website says that there are no funding sources, yet the paper lists 2. Please reconcile.

We have updated the funding information in the manuscript submission system.

Reviewer #3:

This is a very interesting paper that introduces the concept of an X-ray stimulated emission tomography, where the incoming X-rays ($>1.022\text{MeV}$) stimulate electron/positron pair-production which in turn produces photon-pairs when the positron annihilates with an electron in the medium.

This pair-production tomography (P2T) seems very similar to PET but with a different (and more controllable) source of positrons. However, P2T which is an emission tomography seems to be constantly compared with CT, a transmission tomography method, throughout the paper.

PET is physically different from CT and P2T, yet related in detection and reconstruction methods. Positron emission in PET originates from radioactive decay of the tracers, whereas the P2T positrons arise from X-ray interactions with matters. CT detection relies on X-ray interaction with matters in a similar way. As a result, the PET signals are proportional to the amount of radiotracer uptake, which is determined by physiological activities such as metabolic process and blood flow. On the other hand, both CT and P2T signals are determined by the material composition through physical processes. Therefore, P2T is more similar to CT, and the comparison of CT and P2T better demonstrates the uniqueness and strengths of P2T.

P2T is explored using Monte-Carlo simulation through the Geant4 software. Three different image formation processes (each with different pros/cons) are compared and evaluated.

The main question that arises would appear to be: "wouldn't that method have a very large and high-energy dose on the patient?" This does not really seem to have been addressed adequately or explicitly.

The P2T imaging applications (Figure 2-4) have a similar imaging dose to CT. We added the dose information to the manuscript for each specific case. We think the imaging application with an acceptable dose can also be interesting to potential readers.

A very compelling case that nullifies this question is the ability to image the dose when performing radiation therapy. This idea alone is sufficient for a great paper.

You could consider shortening the title to "Pair-production tomography" or add a bit more description: "Pair-production emission tomography"?

We shortened the title as suggested.

Have you considered doing transmission CT from the attenuation of the primary beam at the same time?

Yes. In principle, we can acquire Mega-Voltage CT projections without additional irradiation/dose.

While I think this is some really great work, I'm not sure that the ideas are sufficiently developed for nature/BME. This is foundational work, but I think perhaps the first physical demonstration of PPT would more merit publication in nBME. Physical realisation of PPT would require many problems/limitations to be overcome that are completely missed by Monte-Carlo simulation.

Experimental demonstration of P2T would require a substantial hardware setup that is beyond the capacity of our lab that focuses on computational research. On the other hand, publication of the manuscript will inspire well-equipped labs to perform the experiments in a much shorter timeframe. It is worth noting that the second reviewer pointed us to a 1998 paper by Tavora et al. using the same physics for material detection. The experimental study substantiates the physical feasibility of P2T.

Some specific major comments:

line 11-12: "We studied three P2T acquisition methods, including filtered back projection (FBP), time-of-flight (TOF), and scanning pencil beam (SBP)."

Filtered backprojection is not an acquisition scheme, it is a reconstruction method.

The methods should perhaps be described as something like:

"We studied three P2T acquisition methods: a scanning pencil beam (SBP) and two full-field illumination methods, one with temporal resolution sufficient to identify photon-pairs and one with higher temporal resolution sufficient for photon time-of-flight (TOF) estimation."

Done

line 15-16: "...the ability to form tomography with as few as a single heavily truncated X-ray beam," This sentence does not really make sense without reading the whole paper; even then, heavily truncated is a vague term - what is truncated? Perhaps you could say something like: "PPT is a form of emission tomography, so even a single direction of illuminating X-rays are sufficient for tomography and the beam can illuminate a region-of-interest (ROI) only."

We changed it to:

"the ability to perform tomography with just one highly-truncated x-ray beam through the region-of-interest (ROI)"

line 25-26: What makes PET expensive? Wouldn't PPT be similar in expense? (unless it is the production of tracers that's the expensive part I guess?)

In the original submission, we mentioned that X-ray systems are inexpensive compared with PET. PET scanners cost more than CT. The average PET scan time is also longer, making it more expensive than CT.

The tracer is an extra cost that P2T does not require. However, since the cost is irrelevant to the current manuscript, we removed the corresponding reference in the revised manuscript.

line 38-39: "As a result, it is difficult to obtain clean material information from the X-ray CT images."Should possibly mention dual-energy CT (DECT) as a solution for this?

We added the following text:

"Dual-energy or photon-counting CT help improve material differentiation, but they are limited in the number of differentiable basis materials and do not directly quantify material atomic numbers".

line 39-41: Another weakness of CT is poor soft-tissue contrast due to the similarity in densities and the diminishing photoelectric effect with low atomic number materials. "Should possibly mention phase-contrast methods as a solution for this?"

We agree with the reviewer that phase-contrast CT is a potential modality to provide much improved soft-tissue contrast. However, due to the stringent coherent photon requirement, it is not a general-purpose imaging modality like CT or MR. We prefer not to include it in the introduction for conciseness but if the reviewer feels strongly, we are happy to add a paragraph.

lines 42-49: talk about region-of-interest problem in CT. This is only relevant for transmission tomography; PPT is an emission tomographic method, like PET and SPECT. Perhaps should talk about invention/developments and applications of medical emission X-ray CT here as well, i.e., PET and SPECT, which seems more relevant to PPT.

In the context of external beam source, imaging dose, and limited ROI, P2T is more comparable to CT. Therefore, we focused on CT in the introduction. We mentioned PET when pair detection is described throughout the manuscript.

line 50-52: "The ionizing radiation from high-energy X-rays can break DNA strands, which, if not repaired, leads to cell death."This seems serious, why would I choose to do PPT then, unless measuring dose from radiotherapy? This needs to be addressed later if PPT is being proposed outside of a dose measurement application.

Ionizing radiation takes place in both CT and P2T imaging. The P2T imaging applications in this study use a dose comparable to CT scans.

line 84-85: "...the positron loses its kinetic energy and comes to a near stop, it comes into contact with an electron with nearly simultaneous annihilation..."What is a typical distance that the positron travels before annihilation? I assume that it's a very small distance in water... however, this would be a lower limit to PPT resolution right?

The median of positron travelling distances before annihilation is 4.6 mm, and the median of initial positron kinetic energy is 1.1 MeV. The following figure shows histograms of the distances and energies:

Yes this would be the theoretical image resolution of reconstructed P2T images without post-processing. On the other hand, with a known source energy spectrum, the positron traveling distribution can be simulated. A higher resolution P2T image can be resolved through deconvolution with the known kernel.

lines 124-134: Describes the tomographic image generation process for the three methods. This is not quite clear, needs a bit of expansion (particularly the TOF method).

We changed it to:

“For detectors with a high time resolution (e.g., 20 ps), the range of the annihilation event along the LOR can be narrowed down (to, e.g., 3 mm), according to the time difference of the two photons (The TOF method).”

It seems that the first method that uses FBP is a conventional CT-type method while the TOF and SPB methods are more direct imaging techniques, i.e., not inverting Radon transform, rather just adding data to the image as it is measured. Why did you not add some regularisation to the FBP method, e.g., TV-minimisation?

We presented a comparison of TV with FBP in the comment to Reviewer #2. The noise suppression by TV did not offer better conspicuity.

line 131-133: "For detectors with a high time resolution (e.g., 20 ps), the range of the annihilation event along the LOR can be narrowed down according to the time-of-flight (TOF) of the two photons." 20 picosecond resolution means that pair-production position can be determined to about $0.3 \times 10^{12} \text{ mm/s} \times 20 \times 10^{-12} \text{ s} = 6 \text{ mm}$ resolution. So this is the lower limit to resolution here? Should this be mentioned? To improve resolution, could this TOF data be reconstructed using the FBP method without timestamps as well as using the TOF information in a statistical sense?

The lower limit of resolution is $0.3 \times 10^{12} \text{ mm/s} \times 20 \times 10^{-12} \text{ s} / 2 = 3 \text{ mm}$. Note that the number needs to be divided by two because 1 mm offset will result in 2 mm difference in travel between the two photons. Using TOF in a statistical sense could improve image quality, but beyond the current scope.

lines 135-139: Corrections for fluence variation outlined. Where do the "radiological pathlengths" come from? are they pre-measured by CT? does this mean that true ROI is not really possible?

The radiological pathlengths were computed from CT. They are insensitive to fine structures and therefore even ultra-low-dose CT could provide a good estimate on the correction coefficients.

Mention that this is explained in detail later in the paper (lines 417-439)

Done

Table 1: Would be good to spell out acronyms in the caption. It took me a while to work out that GT was ground truth.

Done

line 154-155: "Therefore, in theory, P2T image contrast also follows a simple linear relationship with ρZ ."

Mention that this is explained in detail later in the paper (lines 440-456)

Done

line 159-162: "The convolution of poly-energetic X-rays with non-linear cross-intersections inevitably renders multiple material differentiation tasks an underdetermined problem." Mention that it is common to take additional information using a second X-ray spectrum (i.e., dual-energy CT) for this reason. If PPT is related to ρ and Z , isn't this also underdetermined? can't distinguish between something low density with high Z from something with high density and low Z ?

We added a discussion on DECT to the introduction.

If ρ is known, then PPT signals only depend on the atomic number Z . This is fundamentally different from the material decomposition problem in CT, where the relationship between Z and image intensity is complicated by the convolution of the source spectrum and different physical interactions.

The analysis graphs presented with the images in Figs 2-4 are different every time. This makes it difficult to compare/contrast. Please make them all consistent (preferably the same as in Fig. 2 using a single chart)

We emphasize different properties of PPT in Figs 2-4. In Fig 2, we use scatter plot to demonstrate the linear relationship. In Fig 3, we compare the contrast of PPT with CT. The barplots better present these contrast differences. Besides, the materials are different in the two figures, therefore their image contrasts are not comparable. In Fig 4, we want to compare partial-view with full-view to show their similarities. Again, the barplots better demonstrate the comparison.

line 208-209: "The ρZ factor offers greater contrast for materials including the lung inhale, lung exhale, adipose, and breast tissue." I find it hard to tell for some of these regions. Can you point the reader to specific things to look at to observe this?

We pointed out that "the corresponding rods indicated by the red arrows are more visible in P2T images" for these imaging regions.

lines 220-225: Again, this is comparing an emission CT (PPT) with transmission CT - not really worth doing? Maybe should instead compare PET and SPECT; similar but PPT can be in a targeted region --- could compare radiation doses for PET/SPECT/PPT?

As discussed, other than the shared pair detecting method, P2T are closer to CT than PET and SPECT.

lines 226-245: did these simulations use fluence correction? Measured using CT? this should be mentioned...

The following text was added to the manuscript:

"Note that the fluence correction step in P2T reconstruction still requires the X-ray attenuation coefficients of the entire patient volume, but the fluence is insensitive to minor structures. An ultra-low-dose CT scan could be used repeatedly for fluence correction of many P2T acquisitions on the same imaging object."

line 234-235: "In the specific case, the maximal imaging dose is around 3.3 cGy, assuming 100% detector efficiency."

Please explain how detector efficiency is related to imaging dose?

When the detector efficiency is low, we need a larger fluence to collect the same amount of detector signals (to achieve the same signal-to-noise ratio), which increases the imaging dose.

line 254: please explain what "energy deposition kernels" are and how they are determined.

We changed it to:

"The radiation dose is the local energy deposition from both primary photons and secondary particles. The dose can be computed either through Monte Carlo simulation or by convolving TERMA with energy deposition kernels to account for the energy spread due to finite traveling of secondary particles. We use Monte Carlo simulation to compute dose in this study."

The dose deposition kernels were determined using Monte Carlo. We explained it in the method section.

"The Monte-Carlo precomputed energy deposition kernels account for the energy spread due to finite traveling of secondary particles."

line 256-258: "The P2T images, defined as the number of annihilation events, show the number of pair production interactions convoluted with the positron traveling."

Two questions:

1) Does the "energy" of each pair-production event need to be accumulated rather than the "count" to become more similar to TERMA? or is it sufficient just to scale by 1.022MeV per count?

The accumulated positron energy is a more similar concept to TERMA, as both of them are accumulated energies. However, there is no existing method to directly measure the positron energies in pair production interactions in the interior of the imaging object. On the other hand, all three quantities are proportional to the x-ray fluence. Mathematically, the P2T signals have a similar correlation to TERMA.

2) Should this be something like "convolved with a kernel representing the statistical probabilities of positron interaction within the medium"

We changed it to:

"The P2T images, defined as the number of annihilation events, show the number of pair production interactions convoluted with a kernel representing the statistical probabilities of positron traveling before annihilation within the medium."

line 267: Please explain what "conformal dose" is?

We changed the text to:

"the radiation dose distribution was optimized to achieve prescription dose within the target volume and minimal dose to the surrounding normal tissues, using convex optimization algorithms."

line 270-272: "Assuming the detector efficiency is 10% to collect a coincident photon pair, we simulated 10% of the particles needed to deliver a 2Gy fraction treatment, which amounts to a total of 221.2 billion primary particles." I don't understand this paragraph; why do you need to assume a detector efficiency? can't you model a detector properly? If the detector is 10% efficient, wouldn't you need to deliver 10x the dose, not 10% of the dose?

If the detector efficiency is 10%, only 10% of the coincident photon pairs will be collected as data. We only modeled these collected photons in our simulation. Comprehensive modeling of a specific detector would warrant a separate paper and is beyond the scope of this study. The 10% efficiency we assume in the study is conservative and should set a lower performance bound.

Some specific minor comments:

line 23: "There is a long history of using X-rays for detection." Detection of what? probably need a reference or multiple references here...

Done

line 27-28: "A major breakthrough in X-ray imaging is the invention of tomographic images." Probably need to reference Hounsfield or Cormack here...

Done

line 28: "By acquiring the 1D linear or 2D planar projection images..."

This probably needs a bit more explanation, what are projection images? how can I have a 1D linear image?

The projection images are the attenuated X-ray photon maps after the initial X-ray beams passing through the imaging object. The 1st generation CT acquires 1D projections, and modern 3rd or 4th generation CT machine acquire 2D projections.

We changed the text to:

“By acquiring 2D X-ray attenuation images from many different angles around the patient, a 3D Computed Tomography (CT) image can be reconstructed”.

line 32-33: "Since CT's invention, many technological developments in the hardware and reconstruction methods have significantly improved its speed, image quality, and versatility." Probably need a reference here

Done

line 97-99: "The 511 keV photons consist 0.91%, 15.4%, and 77.6% of the total photons, before applying filters (Figure 1c), after applying an $\pm 10\%$ energy window filter (Figure 1d), and after applying the $\pm 10\%$ energy and 1 ns coincidence time filters (Figure 1e), respectively."

This seems like a confusing way to present the info; "respectively only applies when two items are listed. Perhaps just say: "The 511 keV photons consist 0.91% of the total photons, before applying filters (Figure 1c), 15.4% after applying a $\pm 10\%$ energy window filter (Figure 1d), and 77.6% after applying the $\pm 10\%$ energy as well as a 1 ns coincidence time filters (Figure 1e)."

Done

line 162: missing a "with" before 10

Done

line 177-180: "For example, although the Gadolinium has a lower atomic number than Ytterbium, Tantalum, Gold, and Bismuth, its K-edge energy at 50 keV is closer to the peak of the 120 kVp CT spectrum. Consequently, the CT contrast of Gadolinium is substantially higher than other materials." Excellent description!

line 263: "The dose calculation simulated 10 particles within each pencil beam." Should "particles" be "X-ray photons"?

Yes. We clarified in the manuscript.

References cited in the response letter

1. Mazoyer, B. M., Roos, M. S. & Huesman, R. H. Dead time correction and counting statistics for positron tomography. *Phys. Med. Biol.* **30**, 385–399 (1985).
2. Consuegra, D. *et al.* Simulation study to improve the performance of a whole-body PbF(2) Cherenkov TOF-PET scanner. *Phys. Med. Biol.* **65**, 55013 (2020).
3. Dolenc, R., Korpar, S., Križan, P., Pestotnik, R. & Verdel, N. The Performance of Silicon Photomultipliers in Cherenkov TOF PET. *IEEE Trans. Nucl. Sci.* **63**, 2478–2481 (2016).
4. Lecoq, P. *et al.* Roadmap toward the 10 ps time-of-flight PET challenge. *Physics in Medicine and Biology* (2020). doi:10.1088/1361-6560/ab9500

5. Li, T. *et al.* Dosimetric Performance and Planning/Delivery Efficiency of a Dual-Layer Stacked and Staggered MLC on Treating Multiple Small Targets: A Planning Study Based on Single-Isocenter Multi-Target Stereotactic Radiosurgery (SRS) to Brain Metastases . *Frontiers in Oncology* **9**, 7 (2019).
6. Liu, Y., Shi, C., Tynan, P., Papanikolaou, N. & Papanikolaou, N. Dosimetric characteristics of dual-layer multileaf collimation for small-field and intensity-modulated radiation therapy applications. *J. Appl. Clin. Med. Phys.* **9**, 2709 (2008).

Rebuttal 2

Dear Editor and Reviewer,

We thank you for your revision comments, based on which we have further revised the manuscript. Your comments are individually responded to as follows with corresponding highlighted changes in the revision. We hope that the revision will be found satisfactory.

Sincerely yours

Authors.

Referee #1:

The authors addressed all my comments and concerns, and I'm happy to recommend the manuscript for publication. However, I still do have a few last comments that should be addressed prior the publication.

In the very first intro sentence, my suggestion would be use "imaging" instead of "detection". The latter is too broad and it really does not make too much sense.

Done

Please state clearly that the ToF method uses the SPB approach.

We compared three different methods: FBP, TOF, and SPB. The TOF method does not use or rely on the SPB approach. Instead, TOF uses a high time resolution detector to pinpoint the annihilation event.

We clarified in the abstract:

"We studied three P2T acquisition methods: a scanning pencil beam (SPB) and two volume excitation methods using broad beams. Among the two volume excitation methods, one is equipped with detector temporal resolution sufficient to identify coincident photon-pairs and is reconstructed using Filtered Back Projection (FBP), and the other one is equipped with higher temporal resolution sufficient for photon time-of-flight (TOF) estimation."

Please clarify the range of angles of the P2T 10MV beams for experiments in Fig 2 and 3, similarly to how you describe it for Fig.4.

We clarified that "P2T MC simulation utilized a total of 56.6 billion primary particles in 20 equally distributed coplanar fan beams with full coverage of the phantom in each beam. For SPB, the pencil beam size is $0.2 \times 0.2 \text{ cm}^2$. The pencil beams are excited sequentially, and together all pencil beams cover the entire phantom at each of the 20 beam angles."

You describe the complex dependency of CT contrast formation (Z^3 , shell energy edges..) as a problem, but in my opinion it is rather a good feature, as it greatly enhances image contrast. Please tame down the presentation of the complex CT interactions being a problem.

We certainly did not intend to downplay the value of photoelectric in CT. P2T is complementary to conventional CT in the following two aspects. First, the image contrast has a clean linear relationship with Z which facilitates material decomposition. Second, P2T has higher contrast than CT for low- Z materials such as breast and adipose, as shown in Figure 3.

We clarified in the manuscript:

“..., which give CT excellent sensitivity to materials with mid to high-atomic numbers. On the other hand, the convolution of poly-energetic X-rays with non-linear cross-intersections inevitably renders multiple material differentiation tasks underdetermined.”

Referee #2:

Reading the responses to all of the referees, my overall enthusiasm is considerably decreased as it is now apparent to me that the authors are suggesting a proposed utility as a primary imaging modality (what they refer to as the low dose imaging scenarios). I find that to be ill-informed of clinical realities and to not fully appreciate the value of the “high dose” radiation therapy dose monitoring application, which I found to be extremely exciting.

We are glad that the reviewer recognizes the value of P2T for radiotherapy dose monitoring. At the same time, we apologize if the previous response has caused confusion in the intended applications. The main purpose of the paper is to describe a new method for imaging, dose verification and hybrid purposes, including image-guided radiation therapy. The different applications are not mutually exclusive to each other. We are hoping that the readers will find applications of this technology for their specific needs. In the second revision, we better balanced the different applications and avoided suggesting that P2T is only intended for “purely imaging” applications. We changed the beginning of the abstract to: “To develop a novel modality for real-time radiotherapy dose monitoring and atomic number imaging based on a new image formation mechanism, we introduce pair production tomography (P2T) imaging.”

At the beginning of the discussion, we wrote that “We reported a completely novel X-ray tomography method, P2T, for imaging and radiotherapy dose verification.” We then added that “First, P2T intensities are closely related to dose for in vivo dosimetry. Unlike X-ray induced radiation acoustic imaging⁴³ or Cerenkov imaging²⁵, in-vivo dosimetry using P2T is not limited by anatomical locations and acoustic boundaries.”. We then clarified the potential role for P2T as a complement to CT instead of replacing CT.

I lack enthusiasm for any purely imaging application due to the following:

1. The imaging method depends on a source of photons with energies > 1.022 meV. The broad use of CT in all medical setting requires efficiency in terms of cost of equipment, cost of facility requirements such as shielding and power, cost of space requirements, room throughput, service expenses, and more. Linacs are expensive, require expensive shielding and building footing, take up lots of space, and require service and calibrations not required of x-ray tube based imaging systems. The cost for any linac-based

“imaging system” is a non-starter. This technique only makes sense in the presence of an existing MeV x-ray source.

We acknowledge the lack of immediate access to MV beams in the diagnostic imaging domain. However, I invite the reviewer to consider the potential of starting P2T using some of the over 6000 linear accelerators in the US alone and 14,000 linacs worldwide for dose verification, image-guided radiotherapy, and outcome assessment. We agree that they are not considered “purely imaging” procedures but still provide unique information regarding the low Z material and the radiation dose distribution. We further invite the reviewers to consider thousands of linear accelerators existing in industrial settings for inspection where material differentiation is critical. The latter topic is beyond the scope of NBME, so we would not be able to elaborate in the manuscript. To echo the reviewer’s sentiment and provide a more clear boundary of the current study, we add the following paragraph in the discussion: “We note that MV X-rays produced by linear accelerators are not readily available in a typical diagnostic department for general imaging applications. Therefore, early development of the P2T will likely start at radiation oncology as a means for dose verification and image-guided radiation therapy. The role of P2T may later expand to the imaging realm as a complementing technology to CT with future integration of therapy and diagnosis.”

2. Outside of the radiation therapy environment, the ability to resolve material atomic number adds value to diagnostic utility of CT scanning for only some specific situations. For those situations (e.g., separating iodine and bone), dual-energy CT and photon-counting-detector CT are capable of addressing most of these needs. Where they are limited, such as in evaluation of tendons and ligaments, MRI provides the needed capabilities. The primary value of CT is in fast, robust, high-spatial resolution imaging of anatomic detail. The inherent spatial resolution limits noted in the responses (e.g., 3 mm in the best case of 20 ps resolution detectors for TOF image reconstruction, 45 mm in the case of SPB, which is more practical with modern technology) is already far inferior to CT, which is below 150 micron on the newest high-end CT systems and at or below 1 mm in even low-end systems, which can cost under \$200,000.

As indicated in the first response, we modified the tone to reflect the reviewer’s points and emphasized more its role in radiotherapy. The potential value of P2T is as a complementing modality to standard modalities, not a replacement. We would like to clarify a couple of points in response to this question. First, P2T resolution is not hard-limited by the TOF time resolution. In the case of SPB, the resolution of P2T can be improved by the pencil beam size, which reaches 2.5 mm using existing radiotherapy machines equipped with a high-resolution multi-leaf collimator. Second, we agree that MR is well-suited for soft tissue imaging but does not quantify Z. Therefore, there could still be a role for P2T here.

As a technology not even reaching its infancy, it may be a little harsh to compare every aspect of P2T with mature modalities such as CT that has over 50 years of development under its belt. It may be prudent to save the conclusion of P2T for later, giving P2T an opportunity for future hardware and software optimization. The clinical applications will become clearer with its availability for human studies.

The utility of P2T for imaging effective atomic number outside of radiation therapy treatment planning for proton therapy or radiation therapy dose monitoring pales in comparison to the overwhelming value of the sharp definition of anatomic detail. The blurriness of the P2T images in Figure 3 compared to the sharpness of the CT image already points to this weakness of P2T – and this was in an ideal environment with a point source and ignoring non-ideal detector properties. Also very concerning is the information

provided in the response to reviewer #3 regarding the range of uncertainty due to positron travel: “The median of positron travelling distances before annihilation is 4.6 mm.” The tail of the distribution given extends out to 25 mm. While the authors state that deconvolution with a simulated distribution of positron travel distances can improve resolution, it is not clear by how much and at what expense in noise (deconvolution increases image noise). With such fundamental limitations on spatial resolution, the idea of P2T serving as an imaging modality seems to me a solution in search of a problem. The clear problem that it can solve is in the radiation therapy context and this needs to be recognized and made clear in the paper.

We assumed ideal source and detector models for both P2T and CT simulations. A direct deconvolution will increase image noise, but the noise can be better controlled with model-based image restoration (such as an optimization framework with least-square data fidelity and total variation regularization). Furthermore, as mentioned previously, at the hardware level, P2T resolution can be improved with finer scanning pencil beams.

As mentioned, we modified the manuscript to emphasize more on radiotherapy applications. We hope that the current version presents a better balanced technical description of the P2T for both communities.

3. The simulations – again using ideal source geometries and detectors – used a dose of 36 mGy in a 20x24 cm phantom - assuming a 100% efficient detector. That dose is already a factor of more than 3 higher than typical doses using low end scanners. In such a small patient, 12 mGy would be considered at the upper end of clinical doses. With modern technologies, doses can be in the range of 4-6 mGy in the abdomen and even lower in the thorax. Thus, P2T technology, at some future time when 20 ps temporal resolution detectors are possible, would be at a considerable dose disadvantage to x-ray tube based CT, which will continue to evolve as more dose reduction methods are adopted commercially.

There are several methods to further reduce P2T dose. First, the geometry efficiency of the P2T hardware setup in this study is only 0.17%, which can be increased by a factor of 16 if using ring-detectors with a smaller 60cm diameter, and by another factor of 10 if the detector length is 1m in the patient longitudinal direction, consistent with the whole body PET development. Second, with P2T the imaging dose can be limited to only in the region-of-interest, which is a lot harder, if not impossible, to do with attenuation-based CT.

4. The elements used in Figure 2 to show linearity with atomic number have no real significance in medical imaging, with the exception of Iodine and Barium, which are easily distinguishable in medical imaging due to their method of injection or ingestion. The approval of new contrast agents is a billion dollar prospect and thus it is unlikely that these rare earth materials would be added to the very limited menu of contrast agents. Even though approved for use in humans, and extensively studied and improved when safety issues did arise, there is almost no interest in adopting Gadolinium in CT, even for spectral applications. Thus the linearity of signal to Z offers no compelling benefit for purely imaging applications (unrelated to radiation therapy).

As mentioned earlier, the purpose of figure 2 is not just for the development of new contrast but to show the linearity of P2T intensity with varying atomic numbers. The extension of linearity into the high-Z range is particularly interesting for cargo inspection, which is beyond the scope of the current paper but scientifically relevant. Besides Iodine, Barium, and Gadolinium, Tantalum and Ytterbium have been studied for dual-energy double-contrast CT (e.g., PMID: 26892945 and 26356064). Gold and bismuth

have been considered as radioenhancers for more effective tumor-killing (e.g., PMID: 26700713 and 28145723) with ongoing efforts and a clear path towards regulatory approval. We agree with the reviewer that the development of new imaging contrast is not trivial and may not always result in a clinically viable product. To avoid the potential confusion that the current paper is suggesting the clinical availability of these contrast, we added the following in the discussion: “The current application for multiple imaging contrast differentiation is limited by the few elements approved for clinical use, while the others in Figure 2, including Tantalum, Ytterbium, Gold, and Bismuth, are still in the preclinical stage as potential imaging contrast.”

5. In figure 3, none of the P2T techniques differentiate liver from muscle from water – the effective atomic numbers are too similar. Even bone has lower contrast from water in P2T than in CT. The only materials that show improved contrast relative to water are breast tissue and adipose tissue (where there is a small difference in atomic number). The CT simulation is for some reason not a good indication of CT’s ability to image fat. Clinically, fat is extremely easy to differentiate from other forms of soft tissue. I see no benefit in P2T if it can give more contrast to fat, which is already well seen, but has no contrast from muscle or liver, which are also already well seen and are essential to be seen.

We agree with the reviewer that it is easy to differentiate fat and muscle on a CT image with a proper window/level. However, as shown in figure 3b, P2T provides an order of magnitude higher contrast on the breast tissue than CT. Researchers will find additional unique applications with future fully developed P2T systems motivated by this paper, just like many clinical applications that were not foreseeable at the onset of CT.

6. The answer to referee #1’s question about scan time is 28 s, although they indicate this is only an “order of magnitude” estimate. It is not clear how much imaging volume this would include, which presumably would depend on the number of detector rows (and their resolution) longitudinally. Current systems can scan an adult from head to toe in around 20 s. Some systems can do this in under 5 s. Short of whole-body PET systems, which are out of the question from a cost point of view, scan time would be a major issue for P2T.

The 28s scanning time assumed a low dose rate of 0.1Gy/min. Existing therapeutic linacs operate at a typical dose rate of 10Gy/min, which would reduce the scanning time to less than 1s.

Additional comments:

7. The limits to resolution for SPB (45 mm) should be noted, as well as the limits to resolution due to travel of the positron from its point of creation before its annihilation.

The SPB method has a resolution of 2mm (using 2mm beam width). We clarified in the manuscript as follows:

“The time-of-flight (TOF) information of the 20 ps detector allows directly locating an annihilation event with 3 mm resolution. TOF of 300 ps results in a 45 mm range, which in itself is inadequate resolution. However, when a large number of annihilation events are recorded, such as in high-dose radiotherapy, useful images can be reconstructed in a statistical manner using FBP. SPB increases resolution to be 2mm resolution by using a 2 mm excitation pencil beam even with a 300 ps detector. Therefore, we only consider TOF reconstruction for 20 ps detector and SPB or FBP for 300 ps detector in this study.”

8. Better differentiation between “approaches,” “acquisition methods,” and “acquisition and reconstruction methods” are needed. Even after multiple reads, it still gets confusing. An examples is in the legend for Table 1. What is the difference in imaging methods and imaging approaches? I found the phrase “volume imaging” or “full FOV” to be clear (used in Figure 4). Table 1 needs a matrix to clearly show which combinations of irradiation and which reconstruction approaches are investigated. I also don’t understand why attenuation correction is not applicable to SPB.

We changed ‘imaging approaches’ to ‘excitation approaches’ and clarified that the four imaging methods as different ‘Imaging acquisition and reconstruction methods.’

‘volume imaging’ and ‘full FOV’ are distinct terms. We added a panel (figure 1f) in Figure 1 for clarification. ‘volume imaging’ refers to the excitation approach. We changed it to ‘volume excitation’ as a clarification. Different from the SPB excitation approach that excites a pencil beam path at a time, ‘volume excitation’ irradiates the desired regions-of-interest (ROI) simultaneously. The desired ROI can be ‘full-view’ or ‘partial-view’. The ‘full-view’ means that the entire imaging object is irradiated at each imaging beam angle. Both ‘partial-view’ and ‘full-view’ images can be acquired with either SPB or volume excitation.

We added the following text to the manuscript:

“Two P2T excitation approaches are investigated in this study (Figure 1f). The volume excitation approach excites the entire imaging field simultaneously at each view angle. In scanning pencil beam excitation (SPBE), the imaging field is excited sequentially. Note that the imaging field can be full-view or partial-view of the imaging object regardless of the excitation approach. Full-view imaging covers the entire imaging object, and partial-view imaging only irradiates the ROI without exposing the majority of the imaging object.”

9. Lines 11-14 state: We studied three P2T acquisition methods: a scanning pencil beam (SPB) and two full-field illumination methods, one with temporal resolution sufficient to identify coincident photon-pairs and one with higher temporal resolution sufficient for photon time-of flight (TOF) estimation. However, Figure 4 shows results for full and partial FOV, implying that SPB was also studied for full-field illuminations and FBP and TOF also studied with partial view illumination. Please increase the clarity as to which field illumination technique and reconstruction approach were studied, since Figure 4 implies all combinations were but the text gives only specific combinations.

We changed ‘full-field illumination to ‘volume excitation’. SPB is the counterpart of ‘volume excitation’. SPB can be used to acquire either ‘full-view’ or ‘partial-view’ images, depending on if the irradiated regions at each beam angle cover the entire object or not.

10. Lines 76-77: Different from a previous work using pair production for one-sided point material detection, ... please clarify the meaning of one-sided point detection.

We have clarified in the manuscript:

“A previous study used pair production for one-sided point material detection, where the radiation source and a single detector module are located on the same side of the object, and only a single point can be measured at a time. Different from the previous work, we use coincidence information to form a 3D P2T imaging, which significantly expands the capacity for medical applications.”

11. Lines 94-96: An X-ray beam typically used for radiotherapy introduces the pair production photons This is a confusing phrase. Pair production is an interaction, not photons. There are “pair production” electrons and positrons as they result from the interaction. Please differentiate that the photons interact via pair production to create the electron-positron pair. This would be a good place to discuss the issue of positron travel prior to annihilation. The coincident photons are annihilation photons. Please help the reader to keep these straight.

We added the following text:

“An X-ray beam typically used for radiotherapy introduces the pair production electron-positron pair in a subject placed at the center of a ring-detector array. Before producing the two time-coincident 511 keV annihilation photons, the positron would travel for a median distance of 4.6 mm for 10 MV beam (Figure 6 in the supplementary materials). The two annihilation photons travel in opposite directions, captured by two detectors on the ring.”

12. Line 105: Insert the word “imaging” in front of “approaches.” This is an example of where it seems 2 different imaging “approaches” are studied but the abstract talks about three acquisition methods. Confusing. Lines 132-133 talk about three imaging acquisition and reconstruction methods but you are really laying out the 3 reconstruction methods (FBP, SBP, and TOF). Some of the confusion is that SBP is a unique acquisition method that enables a unique reconstruction method. It seems to me that volume or full-field irradiation/illumination (acquisition) is used with FBP and TOF reconstruction, where TOF is only done assuming 20 ps detector but FBP assumes 300 ps detectors (as does SPB). Table 1 should lay out acquisition and reconstruction and detector assumptions separately and you need to stay with the same terminology throughout that clearly defines the combination you are referring to. The terms approaches and methods are not specific, but the terms acquisition and reconstruction are.

We clarified the terminologies throughout the manuscript and table 1. We now use ‘scanning pencil beam excitation (SPBE)’ vs. ‘volume excitation (VE)’ as two different excitation approaches, and ‘full-view’ vs. ‘partial-view’ to describe the imaging field-of-view. The 3 imaging acquisition and reconstruction methods are SPB, FBP, and TOF. Among them, SPB uses SPBE, while FBP and TOF use VE.

13. Line 191: Gadolinium ... its K-edge energy at 50 keV is closer to the peak of the 120 kVp CT spectrum. Consequently, the CT contrast of Gadolinium is substantially higher than other materials. The k-edge at 50 keV is closer to the effective energy of a 120 kVp CT beam (i.e., 70 keV), not the peak energy. I’m not sure this reasoning is correct, however, seeing as the k-edge of Lu is 63 keV and Ta is 67 keV, both are which are closer to 70 keV than 50 keV is. The signal amplitude in CT is not just about attenuation coefficients. The energy response of the detector comes into play. Around Line 180, where the CT simulation information is given, the detector material, density and thickness are not detailed. The use of any collimator grids is not noted (which are essential in CT imaging to decrease scatter and improve contrast resolution). The beam filtration is not given. CT beam filtration is very different than what is used for 120 kVp in radiographic and fluoroscopic applications. Without these key details, the work presented cannot be replicated and the realism of the CT simulation cannot be assessed. All of the MC simulation experimental details should be summarized in a table. Include an appendix if necessary.

We agree with the reviewer that for an actual CT system, many factors come into play. Here we aim to provide a simplified but replicable model for CT acquisition. The effective energy is defined as the monoenergetic photon energy which has the same attenuation to a given material, such as soft tissue. K-edge materials are attenuated differently than soft-tissue – they will be attenuated differently by a

120 kVp X-ray beam and a 70 keV beam (the effective energy). In particular, since the attenuation coefficient is significantly higher at its k-edge energy, the amount of attenuation is dictated by the number of photons at around k-edge energy. A higher number of photons (peak of the energy spectrum) at the k-edge energy will increase attenuation by the most. Therefore we compare the peak energy with the k-edge energy. The peak energy of 120 kVp diagnostic X-rays depends on the choice of filtration but is typically close to 50 keV (PMID: 8290728).

Details of CT simulation were added in the methods:

“We also assumed an ideal geometry for CT. The poly-energetic CT source is modeled as a single point source with 4.3 mm Al filtration, and the detector response is ideal with no crosstalks. The simulation was based on a very thin fan-beam geometry (5mm fan beam), and therefore anti-scatter grid was not simulated. The impacts of non-ideal geometry on image quality can be found in previous literatures⁴⁷⁻⁵⁰.”

14. Line 243: The phrase “ultra-low-dose” is hyperbole. Give a numerical estimate based on the literature. Be aware that CT scanners cannot deliver infinitely low doses. Doses are bounded by realistic tube and generator specifications.

We changed the text to:

“One CT scan with the low dose protocol (i.e., 10% of the full imaging dose, depending on the scanner setting) could serve for fluence correction of repeated P2T acquisitions on the same imaging object.”

15. Line 381: What is “perfect beam shaping?” Beam shaping in what respect? Spectral? Spatial? Point focal spot? Shielding is mentioned. What impact would that have on the results of the simulation? Presumably degradation in spatial resolution due to increased spread of the focal spot?

We clarified that ‘we assumed an ideal point source without leakage’. Note that we assumed the ideal point source for the CT simulation in this study. We expect both CT and P2T images will be more blurred using a real source.

16. Line 385: Electronic noise and detector crosstalk definitely will (not may) impact SNR.

Corrected.

17. Unfortunately, insufficient detail is still a problem.

We included more details on the CT and P2T simulation. We are confident that our simulation should be 100% reproducible by the readers with the given details. However, we are happy to provide more information if the reviewer kindly points out what is missing.

18. The results with the fast detectors (time of flight, TOF) and volume acquisition are very nice, but the practicality of 1440 of these 20 ps detectors is difficult to envision at the current time. What technical advances in detector technology are needed to achieve commercial viability? Authors: Cherenkov light is a potential approach to achieve 20 ps detector time resolution. A simulation study reported an ideal photo-detector time response at 22 ps². But it is difficult to achieve both good detection efficiency and time resolution using Cherenkov light. Only 6% single side detection efficiency was reported³. Achieving high time resolution without sacrificing detector efficiency is required for commercial viability. The

discussion should more clearly emphasize the long road to technically achieve 20 ps resolution as many readers of this journal will not be familiar with the limitations of detector technologies.

We added the following text to the discussion:

“We would like to point out that ultrafast TOF detectors are an active area of research with many technical challenges to balance time resolution and detecting efficiency. In the study, we simulated detectors with a 20 ps time resolution to localize the annihilation event within an accuracy of 3 mm.” The discussion is followed by additional background with references for interested readers to study further *“The 20 ps time resolution detector is consistent with the roadmap of PET detection using prompt Cherenkov emission^{31,32} or ultrafast emitting quantum-confined systems^{45,46}, but both still require significant engineering development to be practical⁴⁷.”*

19. The authors note that 20 ps is consistent with PET detector technology development roadmaps, but state that considerable engineering will be required to achieve this; what timeframe do they think realistic? Authors: 20ps PET TOF detector is still under active research development⁴. Its commercialization is pending upon overcoming many technical difficulties. We do not have enough information to estimate a timeline. Same point of above. Enthusiasm about this paper depends in a large extent to practicality, especially since it aims to offer an alternative to CT, which is already well developed and ubiquitously deployed.

First, as explained, P2T is not meant to replace CT. It has a unique role in radiotherapy for dose verification and a complementary role to CT for local tomography and material quantification. Second, according to Lecoq et al. (PMID: 32434156), 30 ps TOF detectors will be available in the next few years, followed by detectors with 10 ps resolution in the more distant future. Third and most importantly, the fast TOD requirement can be circumvented using SPB with available scanning X-ray pencil beams. SPB provides similar image quality to TOF.

20. The scanning beam approach to data acquisition is suggested as an alternative to requiring extremely high detector temporal response times (e.g., 20 ps). The practicality of scanning photon beam (SPB) treatment protocols should be discussed. What would that look like in terms of extending treatment times? The results are nice, but is this likely to be clinically viable? They note that current linac multi-leaf collimators could be used to perform the pencil beam scanning, but to what extent will that lengthen the treatment time relative to volume imaging. Authors: SPB is viable with current technology. A typical multi-leaf collimator (MLC) has 5mm leaf width (pencil-beam width). The beam width can be further reduced to 2.5mm using the dual-layer MLC^{5,6}. Treatment machines with dual-layer MLC are already commercially available. The total imaging time of SPB using MLC includes beam-on time and leaf-moving time. The SPB beam-on time is comparable to that of volume imaging (20s to 30s), which is limited by the maximum photon flux due to a finite detector dead time. The leaf-moving time is limited by the maximum leaf-moving speed. For example, assuming the maximum leaf-moving speed of 2.5cm/s, it takes 50s leaf-moving time to scan a 5cm-by-5cm imaging field-of-view (2s per row and 25 rows in total). Is the 50 s leaf moving on top of the data acquisition/collection time? This journal has broad readership and the authors cannot assume familiarity with MLC technology. The introduction provided very broad, even somewhat basic, information but in other sections specialty expertise is assumed. Even if 50 s is the total time, that is completely untenable for patient imaging, which much be performed within one breathhold (20 s is an upper limit for most patients). Plus, this covers only a small in-plane field of view (5 cm x 5 cm). What is the coverage in the long axis?

A newer radiotherapy treatment machine (e.g., Varian Halcyon) has a multileaf collimator (MLC) leaf-moving speed of 5cm/s, which reduces the leaf moving time for covering the said area to 25s. It's feasible to achieve a higher leaf moving speed for imaging purposes since imaging only requires a constant moving speed, while radiotherapy treatment requires acceleration and deceleration throughout the treatment. Therefore, the SPB scanning time could be reduced to below one breath-hold with hardware optimized for P2T. We currently only consider single-slice coverage in the longitudinal axis. Faster volumetric P2T imaging in the SPB mode requires electromagnetic scanning X-ray pencil beams. Such a system is also being built for radiotherapy (PMID: 31178058) to cover a large volume with scanning X-ray pencil beams in under 1 second.

21. Please make these points more clear in the manuscript:

a. "For a single annihilation event, with imaging dose in the diagnostic range, TOF of 300 ps results in a 45 mm range, which in itself is inadequate resolution. However, when a large number of annihilation events are recorded in radiotherapy with 2 orders of magnitude greater dose, useful images can be reconstructed in a statistical manner using FBP as shown in Figure 5. "

See the response to comment #7.

b. SPB is readily available using current technology.

We added the following sentence to the discussion:

"On the other hand, the SPB does not require fast TOF detectors and is readily achievable using current technology."

c. In the low dose imaging setting, FBP images with broad beam irradiation are included in the manuscript as a reference. SPB and TOF will likely be the method of choice in the low dose mode. In the high-dose therapeutic mode, however, FBP still results in highly usable images (Figure 5).

We added the following text:

"Currently, the FBP images are only useful in high dose therapeutic mode, and the low-SNR FBP images under low dose P2T acquisition are only included as a reference.On the other hand, the SPB and TOF methods result in significantly higher SNRs compared with the FBP method with the same geometry setup."

d. We presented three imaging applications and one radiotherapy application in this study. The three P2T imaging applications use MeV photons only for imaging and not for simultaneous treatment. --- Please enumerate the clinical applications more clearly. I don't see three of them. Also put the potential clinical value into perspective. Also note that region-of-interest CT ("truncated imaging") has been demonstrated for well over a decade but there is no clinical interest as the applications would be so limited. Part of the value of CT is that it identifies non-expected findings. By covering a large volume, it quickly gives information about the entire body region, and that is one of its key value propositions compared to MRI and US, which both require more focused imaging of targeted regions.

The three imaging applications refer to the three phantom studies using P2T as an imaging modality. P2T can be used for both full-view and partial-view imaging. We agree with the reviewer that truncated imaging has not been particularly important from the diagnostic CT perspective. It is due to two reasons.

First, as the reviewer stated, part of the CT value is to non-expected findings with a large FOV. As explained, P2T is not meant to replace CT, certainly not in the context of identifying non-expected findings. P2T is, however, valuable for repeated imaging of a small ROI. A good example is image-guided radiotherapy that images the tumor and its vicinity on a daily basis. It is also useful for the longitudinal study of a specific area for disease progression and response to treatment. Second, heavily truncated CT (e.g., beamwidth <10% of the patient cross-section) not only results in unusable images but also still delivers imaging dose to the entire cross-section with source revolving around the patient. In comparison, P2T image quality is minimally affected by heavy truncation AND can be done with as few as one excitation beam, which truly limits the imaging dose to the strip of tissue. We added the following sentence in the discussion: “The local tomography with localized imaging dose feature is suited for regional localization and longitudinal observation of specific areas that are particularly relevant for interventional procedures such as radiotherapy.”

e. The three P2T images have the same imaging dose.

We added that *“The three imaging acquisition methods (FBP, TOF, and SPB) are simulated using the same number of particles, and therefore having the same imaging dose.”*

f. However, since the cost is irrelevant to the current manuscript, we removed the corresponding reference in the revised manuscript. – I don’t think cost is irrelevant. Not only are the detectors much more expensive, the need for a megavoltage source is vastly more expensive. This paper demonstrates theoretical potential, and implies to the reader that there may be value in such an imaging paradigm. In the absence of practical pure imaging applications, due to the points discussed above, this is unrealistic. I hope that the authors will come back with a revision that hits a home run with the radiation therapy potential applications and avoids suggesting highly unlikely pure imaging applications.

See the response to comment #1. We hope that it is clear that we are not suggesting P2T as purely imaging modality to replace CT or MR, but a dose verification tool for radiotherapy and a complementary imaging tool for image-guided procedures.

g. If the detector efficiency is 10%, only 10% of the coincident photon pairs will be collected as data. The 10% efficiency we assume in the study is conservative and should set a lower performance bound. The dose that you estimate is based on 100% detector efficiency and here you note 10%, implying to me that your dose estimate is not very realistic.

We clarified in the dose estimation that:

“Note that the estimated imaging dose is inversely proportional to the detector efficiency assuming fixed image SNRs.”

Referee #3:

Overall it seems that the authors have addressed the comments of the reviewers quite thoroughly with many modifications to the paper that make it more clear, precise, and readable. I only have a few minor comments:

* Both reviewers 2 and 3 asked about the quality of the FBP result in Fig. 2. I think perhaps this should be addressed in the paper. The TV-min image could be included in Fig 2, or the fruitlessness of regularisation techniques to improve the result should be mentioned.

We added the following text:

“We implemented three reconstruction methods for P2T. Among them, FBP is the least technically demanding and can be acquired with a regular PET detector and medical linac. Currently, the FBP images are only useful in high dose therapeutic mode, and the low-SNR FBP images under low dose P2T acquisition are only included as a reference. Regularized iterative reconstruction minimally improves the low-dose P2T without SPB or TOF and is omitted from the result. The low dose FBP P2T may be possible if using a different detector setup with higher geometric efficiency, i.e., smaller diameter ring or longer detector array.”

* I think I now understand that you are saying that P2T is similar to CT in terms of WHAT it is imaging. I accept that position. I guess it is most similar to X-ray fluorescence CT (XRF CT). I think of P2T as similar to PET in terms of HOW it is imaging, i.e., they are both emission tomography, requiring correction for reabsorption of the emitted radiation by the sample itself (CT does not require that), they are both measuring the X-rays emitted from positron annihilation (CT is measuring incident X-rays not attenuated by Photoelectric absorption and Compton Scattering), the only difference is the source of positrons; one is from radioactive tracers, one is stimulated by an X-ray beam with energy $> 1.022\text{MeV}$. Perhaps this WHAT/HOW distinction could be made clear in the introduction?

The reviewer is exactly on the point. We added the following to the introduction:

“P2T is similar to PET in terms of how it is imaging: they both measure coincident annihilation photons emitted from positron annihilation. The only difference is the source of positrons: PET introduces positrons with radioactive tracers, and P2T introduces positrons through megavoltage X-rays induced pair productions. P2T is similar to CT in terms of what it is imaging: the image signals both depend on the material compositions. They differ in image formation mechanisms: CT measures the X-ray transmission and P2T measures the pair production signals.”

* Regarding the phase-contrast techniques for enhanced soft tissue differentiation, it is my understanding that preclinical systems have been developed that use talbot-lau grating interferometry to remove the coherence restrictions. I think it is worth mentioning even with just a sentence to show that you are aware of this alternative.

We added the following to the introduction:

“Phase-contrast CT can improve soft-tissue contrasts, but it requires either coherent X-ray sources or additional optical elements such as Talbot-Lau gratings. Due to these additional complexities, significant research and development are still needed to make phase-contrast CT a clinically viable modality¹⁷⁻¹⁹.”

* Should your response/graphs to my question about lines 84-85 on positron travel distance be included in the paper? This seems important...

We apologize for missing the question in the previous revision. Here, we added a sentence in the main section: *‘Before producing the two time-coincident 511 keV annihilation photons, the positron would travel for a median distance of 4.6 mm for 10 MV beam (Figure 6 in the supplementary materials).’* Note that P2T image blurring due to the positron travel can be effectively reduced by using thin pencil scanning beams in SPB.

The figure and discussions are now included in the appendix.

* Should your response to my question about lines 159-162 about ρ/Z be included in the paper? This seems to be an important assumption that ρ is known...

We added a sentence: *'If ρ is known, then the atomic number Z is determined.'*

* My question about line 234-235, was an indirect way to say that you should rephrase/expand the sentence to explain that you are assuming a certain measured signal and, if the detector is 100% efficient, that corresponds to a certain dose... can you please do so?

We added the following sentence:

"Note that the estimated imaging dose is inversely proportional to the detector efficiency assuming fixed image SNRs."